# Optimization Method of Sheet Metal Laser Cutting Process Parameters under Heat Influence

**Yeda Wang [1], Xiaoping Liao [1,\*], Juan Lu [2] and Junyan Ma [1]**

1   Guangxi Key Laboratory of Manufacturing Systems and Advance Manufacturing Technology, Guangxi University, Nanning 530004, China; wyd19990730@126.com (Y.W.); 20050063@gxu.edu.cn (J.M.)
2   Department of Mechanical and Marine Engineering, Beibu Gulf University, Qinzhou 535011, China; lujuan3623366@163.com
\*   Correspondence: xpfeng@gxu.edu.cn

**Abstract:** To address the issues of workpiece distortion and excessive material melting caused by heat accumulation during laser cutting of thin-walled sheet metal components, this paper proposes a segmented optimization method for process parameters in sheet metal laser cutting considering thermal effects. The method focuses on predetermined perforation points and machining paths. Firstly, an innovative temperature prediction model $T_p(r, t)$ is established for the nth perforation point during the cutting process, with a prediction error of less than 10%. Secondly, using the PSO-BP-constructed prediction model for laser cutting quality features and an empirical model for processing efficiency features, a multi-objective model for quality and efficiency is generated. The NSGA II algorithm is employed to solve the objective optimization model and obtain the Pareto front. Next, based on the predicted temperature at the perforation point using the model $T_p(r, t)$, the TOPSIS decision-making method is applied. Different weights for quality and efficiency are set during the cutting stages where the temperature is below the lower threshold and above the upper threshold. Various combinations of machining parameters are selected, and by switching the parameters during the cutting process, the thermal accumulation (i.e., temperature) during processing is controlled within a given range. Finally, the effectiveness of the proposed approach is verified through actual machining experiments.

**Keywords:** laser cutting; optimization of machining parameters; heat transfer; artificial neural network; multi-objective optimization





## 1. Introduction

Sheet metal fabrication is widely utilized in various fields, such as automobile, ships, aerospace, and precision mold manufacturing. With advancements in industrial capabilities and manufacturing standards, higher requirements have been placed on the quality and diversity of sheet metal processing. Traditional cutting methods such as tool cutting and plasma cutting are associated with issues such as tool vibration [1,2] and tool wear and generate significant noise during the machining process. These drawbacks are contrary to the principles of green and sustainable development. In comparison, laser cutting, known for its high precision and efficiency, has become one of the most popular cutting methods in the sheet metal industry [3]. However, laser cutting is a thermal process that introduces heat effects during material cutting, which can degrade the surface quality of sheet metal. Therefore, optimizing this process should consider the influence of accumulated heat in the workpiece.

For thin-walled components [4], the thermal accumulation issue can lead to problems such as distortion and excessive melting during machining. To address the thermal accumulation issue in laser processing, it is common to plan the process before manufacturing the workpiece. Process planning primarily involves the optimization of the processing path and the selection of processing parameters. Researchers have conducted

relevant studies on path planning to reduce thermal accumulation effects. The laser cutting path planning problem refers to finding the most efficient cutting path that minimizes the time required to cut all the parts from the sheet metal [5,6]. From the perspective of part quality, if the distance between two adjacent piercing points in the path is too close, the accumulation of heat between those points can lead to workpiece distortion or excessive material melting, thereby reducing the quality of the workpiece. To mitigate the impact of thermal accumulation on processing quality, some scholars have carried out processing path planning for heat effects. Hajad et al. [7] proposed a simulated annealing algorithm combined with adaptive large neighborhood search (ALNS) to minimize the two-dimensional laser cutting path. The algorithm can extract the cutting contour from a given image and find a near-optimal cutting path in the layout of the cutting contour. However, this paper lacks further substantiation as it only compares with commercial CAM software without providing additional evidence or analysis. Han and Na [8] proposed a laser cutting path optimization study incorporating thermal effects. They modeled the problem as a generalized traveling salesman problem (GTSP) with predefined piercing locations. They used simulated annealing to minimize the movement distance of the laser cutting head and imposed penalties when the temperature of the following selected piercing point exceeded a critical threshold. Levichev et al. [9] reduced unnecessary heat accumulation in laser cutting by three different methods and demonstrated the detection of mass degradation due to heat accumulation through a series of experiments. A similar approach was presented by Dewil et al. [10], who employed a penalty-based ant colony optimization algorithm and used finite difference methods to solve the thermal effects numerically. Kim et al. [11] utilized a heuristic backtracking method to optimize laser cutting paths considering thermal effects. In their study, they employed a micro-genetic algorithm to determine the minimum path distance and assigned penalties to selections that exceeded a critical temperature before evaluating the temperature at the next piercing point. The above-mentioned papers were intended to provide readers with a perspective or approach rather than a detailed demonstration or substantiation. It is important to note that further research and experimentation would be necessary to validate and support the ideas presented in those papers. Makbul Hajad et al. [12] proposed a thermal conduction model and incorporated it into the optimization of cutting paths, with a constraint on the critical radius of the thermal influence zone. They penalized cutting paths that overlapped with the thermal influence zone to minimize heat accumulation within the workpiece. While these methods reduce thermal effects and ensure processing quality, they increase the length of the cutting path to some extent, sacrificing processing efficiency. Additionally, there are still regions of thermal accumulation that cannot be avoided in path planning. As mentioned earlier, researchers such as Makbul Hajad et al. have optimized the cutting path to mitigate the effects of heat accumulation. However, there are still some instances of heat accumulation that cannot be resolved through path planning alone. Figure 1 [12] illustrates this issue, where the red area represents regions with higher temperatures. Processing in these areas may lead to heat accumulation and affect the quality of the machining. Additionally, using a constant speed for cutting across the entire workpiece can result in lower cutting efficiency for points with lower temperatures, as indicated by the thermal map. Therefore, optimizing processing parameters after determining the path can improve processing quality and efficiency in suitable regions.

Under the determined workpiece and tool, the selection of processing parameters primarily focuses on the requirements of the laser cutting process, such as processing quality and efficiency. It involves choosing process parameters that meet the processing requirements, such as laser power, cutting speed, repetition frequency, etc. The selection of process parameters not only affects the processing quality and efficiency but also influences the heat generated during the cutting process. Therefore, it is essential to consider the thermal effects when selecting processing parameters under the determined shortest process path and regulate the thermal effects during processing through different parameter settings. This approach is an effective means to achieve stable processing quality and

efficiency. However, current research on thermal effects in laser cutting mainly focuses on the cutting path, and there is limited research on adjusting processing parameters based on thermal effects.

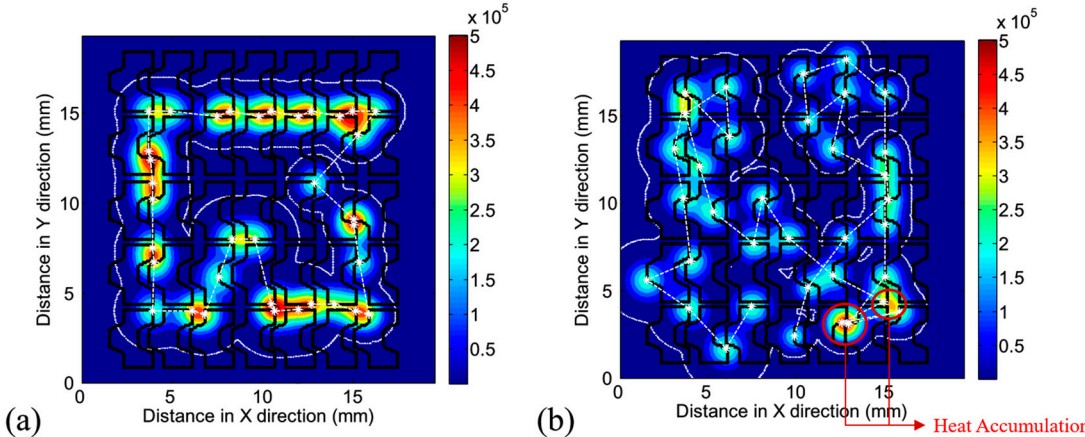

**Figure 1.** Problems in laser processing path planning. (**a**) Thermal maps for laser processing path planning based on thermal effects; (**b**) path planning cannot avoid areas of heat accumulation [12].

Considering thermal effects in processing parameter control under the shortest path primarily involves two aspects of research: Firstly, it consists of constructing a heat transfer model for the cutting process to predict the pre-cutting temperature at the next piercing point. The second part is optimizing the processing quality and efficiency based on heat effects and optimizing processing parameters through multi-target optimization.

At present, there are a few research achievements in the construction of heat transfer models of the cutting process. Yang et al. [13] utilized a finite element model for laser-assisted milling to predict the volume of the heat-affected zone. Their model calculated emissivity and absorptivity based on experimental data and attempted to predict the heat-affected zone under different process conditions. Ju et al. [14] took into account the heat source model, latent heat of phase transition, surface effect element, mesh generation and element generation, etc., and concluded that the maximum temperature of the cladding layer is proportional to the scanning speed of the laser. Gouge et al. [15] improved the convection modeling in laser processing and applied it to thermal simulation, making the model predict the transformation, deformation, and residual stress of the microstructure more accurately. Michaleris [16] analyzed the finite element modeling of heat transfer of metal deposition in laser processing. Pan et al. [17] simulated the scanning of a rotating laser and approximated the absorption ratio based on the molten zone prediction. Nadim et al. [18] studied thermal phenomena during laser irradiation using a finite element model, including the influence of jet cooling and different laser beam power distributions. In most of these researches, the authors focused on the thermal effects during cylindrical and thick plate cutting, with little investigation into the thermal effects and temperature field variations during thin plate processing.

In order to simulate the heat accumulation problem in the machining process, it is necessary to predict the thermal impact of the residual heat of the workpiece on the machining point, and the existing software is not able to make the prediction and then establish the correlation with the simulation optimization framework. Moreover, the currently established heat transfer models have limited capabilities and can only transiently represent the temperature influence of a pierced point on the next unpierced point. These models have limitations as they do not consider the overall thermal influence of residual heat from the entire processing workpiece on the next piercing point. Therefore, in further research, it is necessary to consider the global thermal influence variations on the piercing points and incorporate factors such as heat flow effect into the modeling. Such improvements will help to predict the workpiece state more accurately and enhance the reliability of the model.

In the optimization of laser cutting process parameters, response surface methodology (RSM) and its variants are the most commonly used methods in laser cutting modeling [19–24]. RSM requires specific experimental designs such as Taguchi's method, central composite design (CCD), or Box–Behnken design (BBD). Taguchi's design allows for smaller experimental groups [20,22]. In CCD, the individual effects, square effects, and interaction effects of the factors are estimated more accurately as they provide a better understanding of the endpoints and circumferences, generating a better quadratic model [23]. This method minimizes the number of experiments and assesses quadratic interactions between factors. The use of RSM modeling has the advantages of strong interpretation and relatively low data requirements, but it also has limitations and difficulty in model selection. An alternative and more favorable solution compared to RSM for simulating laser processing is the use of artificial neural networks (ANNs) [25,26]. ANN models do not rely on specific experimental combinations and can achieve high-precision modeling of laser processing based on limited available data from actual processing. Furthermore, multi-output ANN models allow for modeling among multiple quality evaluation metrics and process parameters without the need to establish separate models for each evaluation metric [27,28]. At present, the optimization models that predict the processing parameters in multi-objective optimization of quality consider the entire processing procedure of the part. Typically, a single set of processing parameters is used to process the whole workpiece, such as a sheet. However, due to the thermal effects of laser cutting and their cumulative nature during the cutting process, it is often challenging to simultaneously ensure both quality and efficiency. Therefore, it is necessary to set the appropriate processing parameters based on the different processing environments for each part of the sheet to achieve the maximization of quality and efficiency.

Based on the current situation described above, to achieve process parameter optimization considering thermal effects for the shortest path and to avoid the accumulation of heat that affects cutting quality while improving cutting efficiency, this article starts from three aspects: creation, cutting quality and cutting efficiency of the heat transfer model, and the optimization of cutting parameters. We innovatively proposed a framework for segmented optimization of sheet metal cutting process parameters under thermal effect. The framework aims to achieve stable processing quality under the thermal influence of laser cutting processes while enhancing cutting efficiency.

The remaining sections of this paper are as follows: Section 2 describes the segmentation, optimization, and regulation model based on thermal influence. Section 3 models the heat transfer during the laser cutting process, calculates the heat accumulation at the perforation point, and calculates the heat accumulation from the contour to the perforation point by selecting the feature points. Section 4 designs the experimental design of the orthogonal laser cutting. Section 5 carries out the results and analysis, using a particle swarm optimization algorithm (PSO-BP) to construct a neural network to establish the relationship curve and multi-objective optimization to optimize the process parameters; discusses the optimization results of the process parameters on the basis of the previous model using the Technique for Order Preference by Similarity to Ideal Solution (TOPSIS) to select the parameter combinations to meet the requirements of the temperature regulation by adjusting the parameter combinations; and finally, offers the comparison of the thermograms and experimental verification. Section 6 summarizes the key findings, contributions, and implications of the research and offers suggestions for future work in this area.

## 2. Segmented Optimized Regulation Model Based on Thermal Influence

Compared to traditional mechanical cutting methods, laser cutting faces a significant challenge known as the thermal effect, which can have a detrimental impact on the material being cut. This results in a decrease in the surface quality of the sheet metal, characterized by cutting ripples, dross, burrs, warping, and oxidation discoloration, as shown in Figure 2. In precision machining processes, even slight variations can lead to errors in the dimensional accuracy of the sheet metal parts, directly affecting the overall processing precision. This

paper aims to explore the thermal effects of laser cutting and discuss methods for controlling and mitigating these effects.

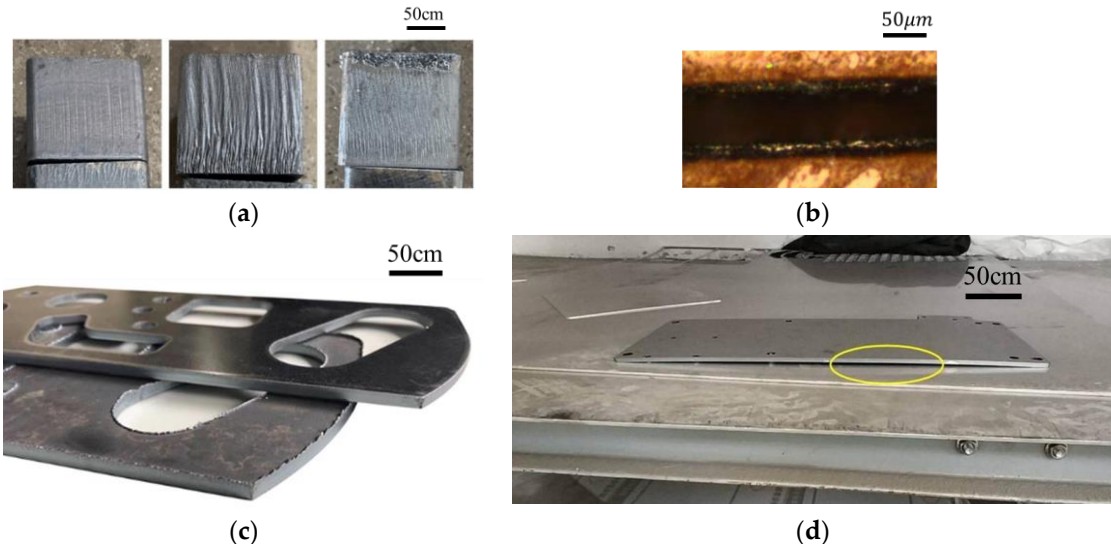

**Figure 2.** Problems in laser processing. (**a**) Cutting ripples; (**b**) dross; (**c**) burrs; (**d**) warping.

This study focuses on investigating the thermal effects of laser cutting of thin-walled components with a thickness of less than 6 mm. A segmented combination optimization method is proposed to control the temperature within a specified threshold range throughout the entire cutting process. The aim is to address the impact of heat accumulation on cutting quality under the shortest cutting path and further improve cutting quality and efficiency. To achieve this, a segmented combination optimization temperature control model, as shown in Figure 3, is proposed. The control principle of the model is as follows: simulate and predict the temperature after each piercing point. When the predicted temperature has not reached the upper threshold $T_1$, the process parameters A are selected for cutting to ensure quality and improve efficiency. As the cutting process progresses, heat accumulates, and when the predicted temperature reaches the set value $T_1$, the process parameters B are selected for processing to improve cutting quality and maintain temperature control below the critical temperature. When the temperature decreases to $T_2$, the processing parameters are switched back to the original combination A for cutting. In summary, the entire cutting process is divided into stages according to the set temperature range. During the stages exceeding the upper threshold, combination A is used for processing to ensure efficiency and quality. During the stages below the lower threshold, combination B is used for processing to improve cutting quality while maintaining temperature control. This approach enables efficient and high-quality processing throughout the cutting process while mitigating the effects of heat accumulation on cutting quality.

The temperature control process based on segmented combination optimization of process parameters can be divided into the following three steps:

(1) Establishing the thermal transfer model (prediction model) for the piercing point and contour in the cutting process to determine the initial temperature $T_0$ at the piercing point and the temperature $T$ after piercing.

(2) Using the PSO-BP algorithm to construct a relationship model between process parameters (processing power, repetition frequency, and cutting speed) and cutting quality characteristics (kerf width and heat-affected zone). The Non-dominated Sorting Genetic Algorithm II (NGSA-II) algorithm is then utilized to solve the multi-objective model for kerf width, heat-affected zone, and processing efficiency, obtaining the Pareto front of the process parameter combinations.

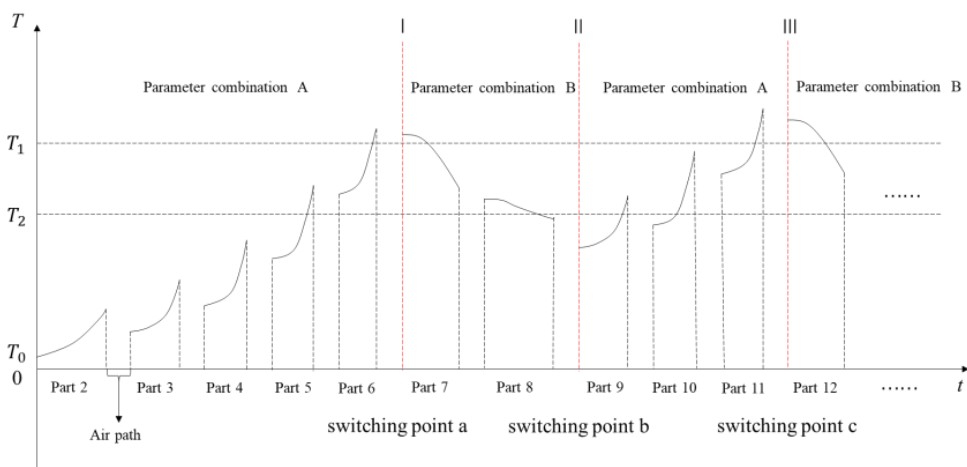

**Figure 3.** Laser processing regulation curve.

(3) Based on the Pareto front, combined with the different weight ratios of the TOPSIS decision-making method and the temperature $T$ after the perforation point predicted by the heat transfer model $T_p(r, t)$, the heat-affected area of the sheet metal cutting process is controlled within a given range. Optimized process parameter combinations A and B are alternately selected based on T to achieve high quality and high efficiency throughout the entire sheet metal cutting process.

Based on the above control process, a comprehensive framework for the process parameter control model based on thermal effects is established, as shown in Figure 4, which includes four parts: establishment of the heat transfer model, data acquisition (design of experiments), modeling and multi-objective optimization, and segmented decision making and validation. Specific steps are as follows:

Step 1: The thermal transfer models $T_n(r, t)$ for the perforation points and $T_{k_n}(r, t)$ for the feature points were established. By selecting feature points on the machined part, an approximate thermal transfer model $T_{C_n}(r, t)$ for the contour heat source was derived. Finally, the thermal transfer model $T_p(r, t)$ for the perforation point n was obtained as $T_p(r, t) = T_n(r, t) + T_{C_n}(r, t)$.

Step 2: A three-factor six-level orthogonal experiment was designed. The temperature, machining quality, and processing time transferred to specific points were measured under different machining parameters.

Step 3: Error analysis was performed on the thermal transfer model. The obtained data were input into the optimized neural network to generate the relationship curves between machining parameters and machining quality and efficiency. The NSGA-II algorithm was used to optimize the machining quality and efficiency, resulting in a Pareto front of multiple objectives.

Step 4: Based on the thermal transfer model established in Step 1, the thermal accumulation under different machining parameters was predicted. The TOPSIS method was employed to obtain decision solutions that meet different objectives by setting and adjusting the weights, ensuring that the thermal impact caused by each machined part remains within a temperature range. The efficiency is improved without sacrificing machining quality. Finally, the conclusions were analyzed through thermal maps and experiment verification.

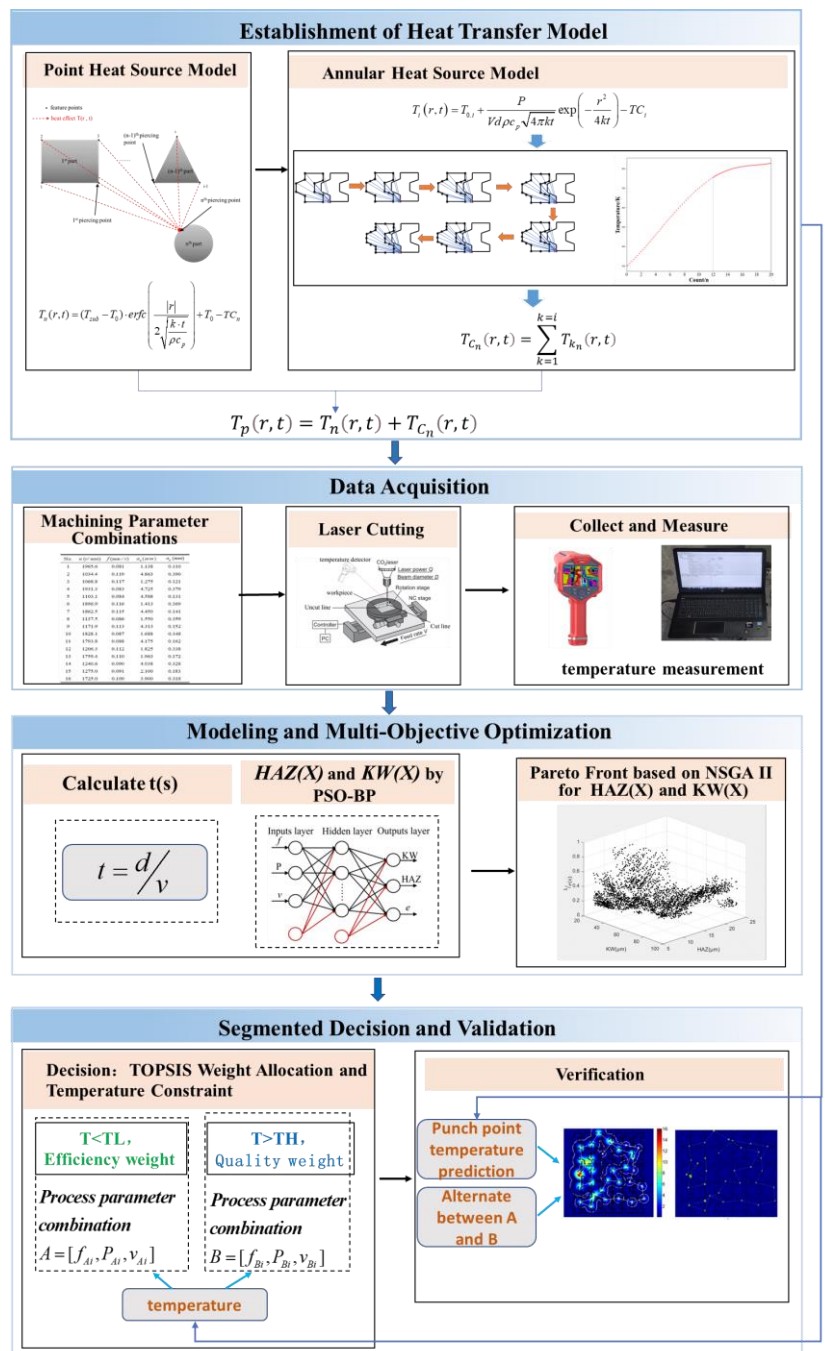

**Figure 4.** A generalized framework for process parameter regulation models based on thermal influences.

## 3. Modeling of Heat Transfer

### 3.1. Generation and Transfer of Laser Cutting Heat

In laser processing of various sheet metal parts with different specifications, the process includes operations such as piercing, contour cutting, and air path. During the piercing and contour cutting processes, heat transfer occurs, leading to thermal effects on the sheet metal. When using a circular laser beam for contour cutting, the boundary of the thermal influence zone is a circle with a critical radius, starting from a single point. After cutting the (n−1)-th contour, the laser beam moves to pierce the nth contour and performs cutting on it. The residual heat left on the surfaces of the 1st, 2nd, ..., (n−1)-th contours will affect the surface of the nth contour. Under unchanged processing parameters, the initial temperature of the surface of the nth contour will be higher than that of the

surfaces of the 1st, 2nd, ..., (n−1)-th contours, as shown in Figure 5. Therefore, as the laser cutting process progresses, the cumulative thermal influence leads to an accumulation of initial temperatures on the contour surfaces. When the temperature exceeds a certain threshold, it affects the cutting quality. Since the heat generated during the piercing process is much greater than that during the cutting process, predicting the temperature at the piercing point before processing can help forecast the temperature at the nth piercing point during the cutting process and specify subsequent process parameter control. This paper establishes a thermal transfer model to calculate the temperature at the piercing point.

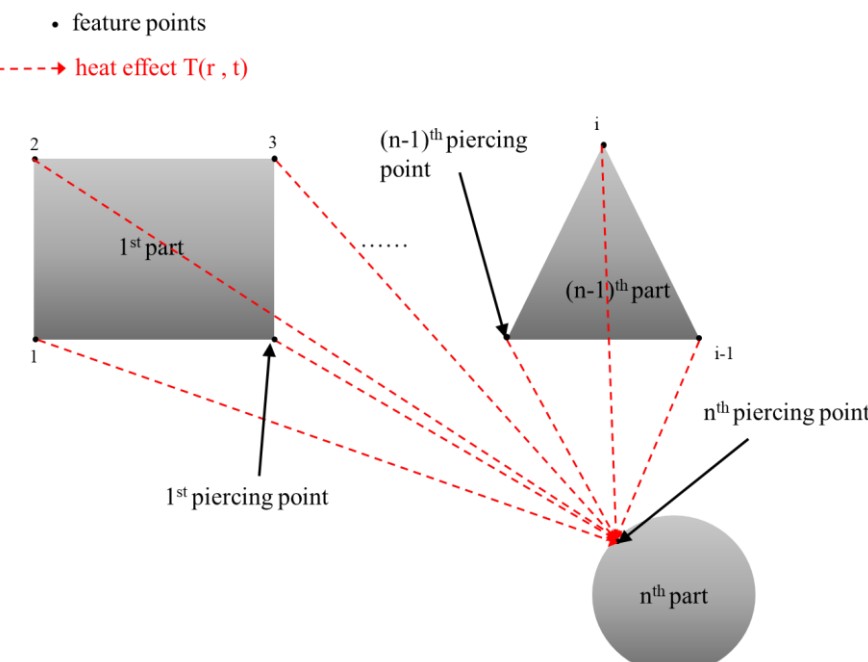

**Figure 5.** Schematic diagram of laser cutting heat transfer.

To simplify the calculation, the contour region of the cutting part is divided, and the heat source is divided into two parts: the heat source from the 1st, 2nd, ..., (n−1)-th piercing points and the heat source generated by the contour after cutting. To calculate the temperature generated by the contour, the contour needs to be divided into characteristic points, and their temperatures are then summed. Therefore, the problem to be solved is divided into two steps.

(1) Heat transfer temperature of feature point k (1, 2, ..., i) to n points → model of $T_{k_n}(r, t)$.

(2) Heat transfer temperature of the actual cutting process contour heat source to position n → $T_{C_n}(r, t) = \sum_{k=1}^{k=i} T_{k_n}(r, t)$.

The schematic diagram of the heat transfer process between the contour piercing point (characteristic point i) and the previously cut contour characteristic points is shown in Figure 5. The established thermal transfer model consists of two parts:

(1) Heat transfer model for 1st, 2nd, ..., (n−1)-th perforation points: $T_n(r, t)$.

(2) Heat transfer model for 1st, 2nd, ..., (n−1)-th profile heat sources: $T_{C_n}(r, t)$.

In summary, the thermal transfer model for its perforation point n is $T_p(r, t) = T_n(r, t) + T_{C_n}(r, t)$.

### 3.2. Heat Transfer Modeling

The heat transfer model refers to the calculation of the thermal effects of the heat sources on position n (piercing point) along a specific process path. The heat sources are divided into piercing point heat sources and annular heat sources. The heat transfer model

for the point heat source is the cumulative thermal effect of the piercing point on the current contour piercing point n, denoted as $T_n(r, t)$. Here, the parameter r represents the distance from the piercing point to point n, indicating that the transfer temperature is influenced by distance. The parameter t represents the duration from the piercing point to point n during processing, indicating the cooling temperature decrease in the processing point after laser machining. To calculate the heat transfer model for the annular heat source, the contour is divided into characteristic points, and the summation of point heat sources forms the annular heat source. Therefore, the transfer heat effect of the actual contour cutting process on point n is denoted as $T_{C_n}(r, t)$. To derive the mathematical model $T_n(r, t)$, we made the following assumptions:

(1) The isotropic materials used in the model have constant optical and thermal properties.
(2) The laser moves at a relatively constant speed.
(3) The laser beam type is a Gaussian beam of constant diameter.
(4) The phase change from solid to gas is a one-step process.
(5) The cut sample is homogeneous and isotropic.
(6) The vaporized material does not interfere with the incident laser beam.

3.2.1. Physical Model of Heat Transfer from the Heat Source at the Perforation Point to the Point at the n Position

Nikita Levichev's transient heat diffusion equation [29] can be simplified to its two-dimensional form as follows:

$$\frac{\partial T}{\partial t} = \alpha \left( \frac{\partial^2 T}{\partial x^2} + \frac{\partial^2 T}{\partial y^2} \right) + \frac{Q}{\rho c_p}, \tag{1}$$

where $t$ is time, $x$ and $y$ are spatial coordinates, $\alpha$, $\rho$, and $c_P$ are the thermal diffusivity, material density, and specific heat capacity of the material, respectively, $T$ is the plate temperature, and $Q$ is the heat input per unit volume.

$$Q = \frac{P \cdot l_c}{v} \tag{2}$$

$P$ is the laser power, $l_c$ is the cutting length, and $v$ is the laser cutting speed.

As the laser beam begins to perforate, the temperature of the workpiece is highest at the perforation point and decays exponentially with radial distance from the perforation. In addition, the thermal influence of the feature points of the previously cut contour accumulates (called temperature buildup) to affect the surface temperature at the perforation point of the current contour. The temperature accumulation is affected by the distance ($r$) from the perforation point and the positioning time ($t$) such that the initial temperature at the nth perforation point ($T_{0,n}$) is expressed as:

$$T_{0,n} = \begin{cases} T_0; n = 1 \\ \sum_{i=2}^{m} T_{i-1}(r_{n-1,m}, t_{n-1,m}); 2 \leq n \leq m \end{cases}, \tag{3}$$

$$r_{n-1,m} = \sqrt{(x_m - x_{n-1})^2 + (y_m - y_{n-1})^2}, \tag{4}$$

where $T_0$ is the initial temperature of the 1st perforation point or the initial temperature of the workpiece (298 K), and n is the total number of perforation points. $r_{n-1,m}$ is the distance from the mth perforation point to the (n−1)-th perforation point, and $t_{n-1,m}$ is the shift time of the laser cutting head from the mth perforation point to the (n−1)-th perforation point.

The expression for the thermal impact caused by laser processing of the perforation point is then

$$T_n(r, t) = (T_{sub} - T_0) \cdot erfc \left( \frac{|r|}{2\sqrt{\frac{k \cdot t}{\rho c_p}}} \right) + T_0, \tag{5}$$

where $T_{sub}$ is the material sublimation temperature and the thermal conductivity of the material is $k$. The complementary error function $erfc(\delta)$ is

$$erfc(\delta) = 1 - erf(\delta) = 1 - \frac{2}{\sqrt{\pi}} \int_0^{\delta} e^{-w^2} dw, \tag{6}$$

where $\delta$ is a parameter in the error function, determined by the specific material, and $w$ is an integral variable in the error function.

Since laser cutting is not performed instantaneously, the temperature of the laser cut contour and the ambient temperature form a temperature potential, so the heat transfer model should also include the cooling of the workpiece during the laser cutting head shift, in addition to the heat accumulation. The cooling rate ($C$) of the alloy is taken from S. Peirovi et al. [30], and the cooling time ($ct_{s,n-1}$) for migrating from the sth perforation point to the (n−1)-th perforation point can be calculated by

$$ct_{s,n-1} = \frac{\sum\limits_{s=0}^{n} r_{s,s+1}}{v}; (s+1) \leq n-1 \tag{7}$$

The temperature reduction at the nth puncture point ($TC_n$) due to the cooling effect is

$$TC_n = C \cdot ct_{n,m} \tag{8}$$

Considering the cooling effect, the temperature field in Equation (5) can be expressed as

$$T_n(r,t) = (T_{sub} - T_0) \cdot erfc\left(\frac{|r|}{2\sqrt{\frac{k \cdot t}{\rho c_p}}}\right) + T_0 - TC_n \tag{9}$$

3.2.2. Physical Modeling of Heat Transfer from a Point Heat Source to an n-Position Point

In the heat transfer model for the same perforation point to n locations, the initial temperature of feature point i in the contour ($T_{0,i}$) is expressed as

$$T_{0,i} = \begin{cases} T_0; i = 1 \\ \sum\limits_{i=2}^{m} T_{i-1}(r_{i-1,n}, t_{i-1,i}); 2 \leq i \leq m \end{cases} \tag{10}$$

$$r_{i-1,m} = \sqrt{(x_m - x_{i-1})^2 + (y_m - y_{i-1})^2} \tag{11}$$

In this study, heat accumulation at the feature points of the cut contour has been used as a constraint for optimizing the machining parameters. To avoid overheating of the workpiece, all combinations of machining parameters beyond the range of the heat-affected zone should be ignored.

Using a two-dimensional transient heat transfer model, the temperature field at the point of the workpiece after perforation can be calculated analytically:

$$T_i(r,t) = T_{0,i} + \frac{P}{Vd\rho c_p \sqrt{4\pi kt}} \exp\left(-\frac{r^2}{4kt}\right), \tag{12}$$

where $P$, $V$, $d$, $\rho$, $c_p$, and $k$ are the laser power, laser cross-cutting speed, laser beam diameter ($d$ = 0.1 mm), density, specific heat capacity, and material thermal diffusivity, respectively.

In order to determine the critical radius of the heat-affected zone due to perforation ($r_{cri,i}$), the derivation is given by

$$r_{cri,i} = \sqrt{4kt \ln\left(\frac{P}{Vd\rho c_p\sqrt{4\pi kt}(T_{cri} - T_{0,i})}\right)},$$ (13)

where $T_{cri}$ is the critical temperature of the heat-affected zone.

Similarly, considering the cooling effect, the temperature field in Equation (13) can be expressed as

$$T_i(r,t) = T_{0,i} + \frac{P}{Vd\rho c_p\sqrt{4\pi kt}}\exp\left(-\frac{r^2}{4kt}\right) - TC_i.$$ (14)

### 3.2.3. Methods for Establishing Contour Heat Source Characterization Points

The heat transfer model of the contour heat source is established by the contour decomposition into feature points. The process of establishing the number of feature points is as follows: establish the heat transfer model of the heat source of the feature points and superimpose the thermal effect of the selected feature points of its contour loop to obtain the temperature generation curve with the number of feature points as the horizontal coordinate. When the temperature tends to be stable or the change is small, e.g., the temperature difference between the temperature of the current feature point and the temperature of the previously selected feature point is less than $\epsilon$ (1% is taken here), then the current feature point can be defined as the temperature model of heat transfer of the contour to the perforation point n.

The feature point selection strategy is as follows:

(1) There are three types of base elements that make up a closed loop: circles, arcs, and line segments; therefore, the connection points between all base elements in a closed loop are specified as feature points.

(2) In addition to the connection points between the base elements, under specific part specifications, line segments are characterized by 2 and 4 equal points; circles are characterized by 4 and 6 equal points on the circle; arcs are characterized by 2, 3, and 4 equal split points on the arc; and irregular contours are characterized by 4, 8, and 12 equal points.

As shown in Figure 6, there are four parts to be machined, which contain five loops. The kth loop is called $Loop_k$ ($k = 1, 2, 3\ldots5$). Each loop has the vertices we have defined earlier, and these feature points are denoted as $V_i$ ($i = 1, 2, 3\ldots20$), where there are four vertices on each of the rings $Loop_1$ and $Loop_2$. The process of feature point generation is to select the points on the connection points, line segments, circles, arcs and irregular profiles to make 2, 3, 4, 6, ... By equal division, the feature points continue to be generated from Figure 6, as shown in Figure 7.

For the processed parts in Figure 1a, two parts are extracted for heat transfer simulation. The feature point generation model is shown in Figure 8. Select the connection points and line segments in the figure to make 2, 3, 4, 6, ... By equal division, the temperature generation curve with feature points as the horizontal coordinate is obtained, as shown in Figure 9. When the temperature difference between the temperature and the previously selected feature points is less than 1%, the temperature shown in the ordinate is the heat transfer temperature of the profile to the perforated point. The general flow chart of heat transfer is shown in Figure 10.

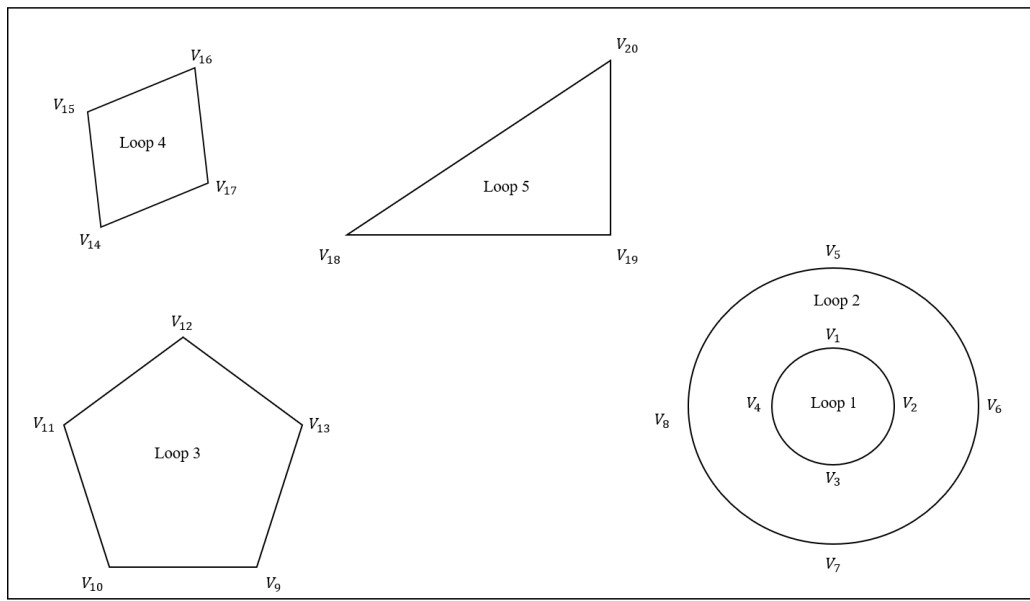

**Figure 6.** Part contour geometry schematic.

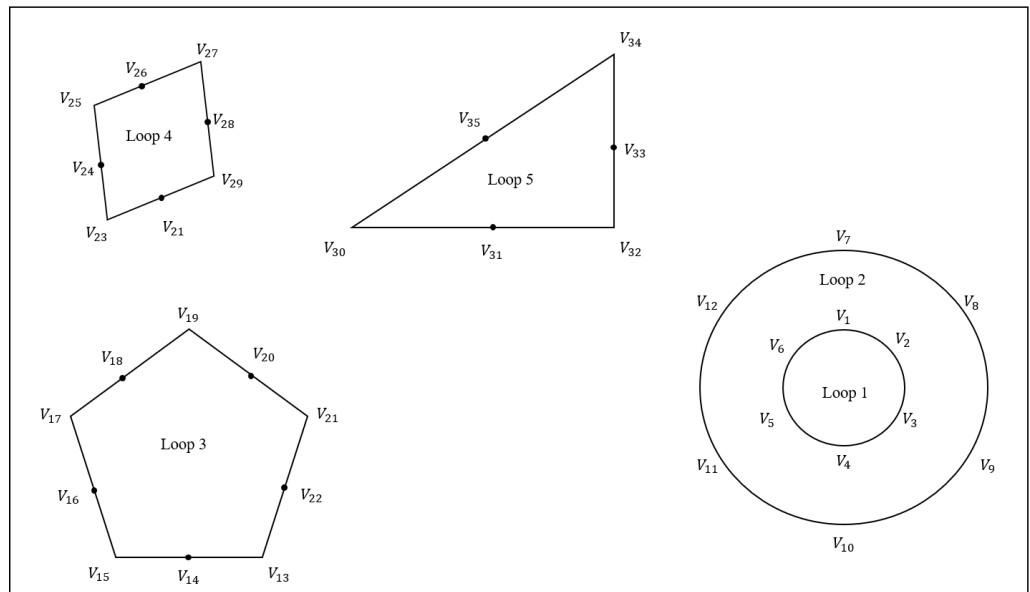

**Figure 7.** Part contour feature point generation schematic.

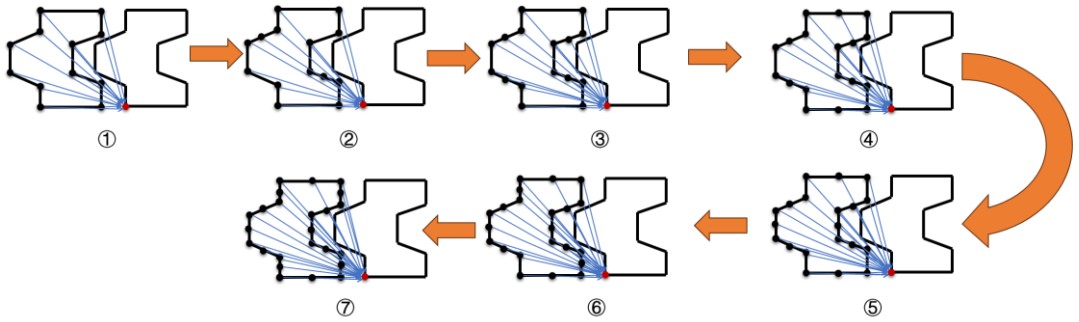

**Figure 8.** Feature point generation model.

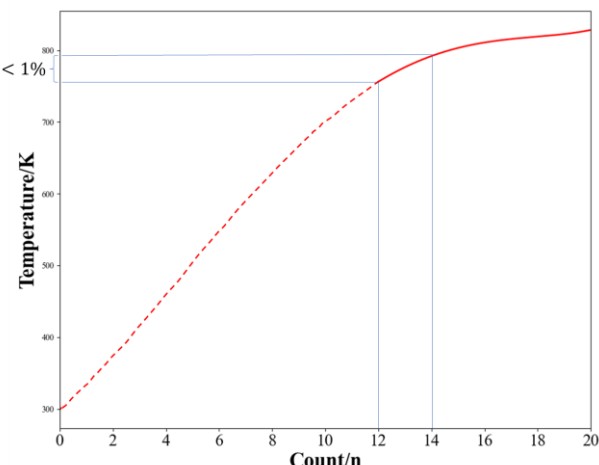

**Figure 9.** Temperature generation curve.

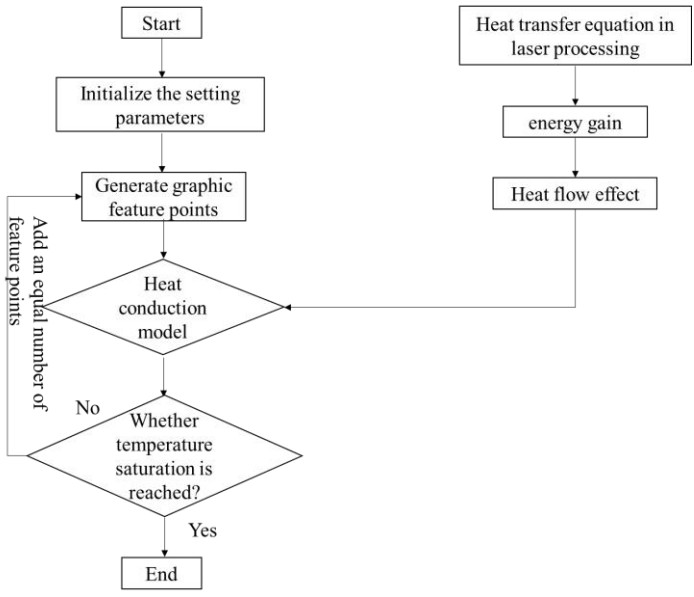

**Figure 10.** Heat transfer flow chart.

## 4. Experimental Design

To validate the effectiveness of the heat transfer model proposed in this paper and provide a data foundation for establishing and solving a multi-objective optimization model, laser cutting experiments were conducted using a 0.6 mm thick low-carbon steel Q195 plate. The experiment aimed to measure the temperature and machining quality of the perforation points under different combinations of cutting parameters. The material properties of the steel plate are shown in Table 1.

**Table 1.** Material properties of steel plate.

| Properties | Value |
|---|---|
| Critical temperature (K) | 995 |
| Density (kg/m$^3$) | 7880 |
| Specific heat capacity (J/kg) | 477 |
| Thermal diffusivity (m$^2$/s) | $1.197 \times 10^{-5}$ |

In general, the quality and efficiency of laser cutting are influenced by process parameters such as laser power, scanning speed, repetition frequency, pulse duration, auxiliary gas

type and pressure, and workpiece type and thickness. Therefore, in the experiment, the repetition frequency, average power, and scanning speed were chosen as process parameters. Based on the machine tool parameters and the laser processing limits of the workpiece, a range of values for the process parameters was set, and six levels were established within this range. The parameter range and levels are shown in Table 2. This paper is based on Windows 11, 3.0 Ghz, 8 G RAM, and python3.9 development environment. Using a custom optimization design, 50 sets of process parameter combinations were obtained, as shown in Table 3. The kerf width (KW) and heat-affected zone (HAZ) were selected as features for cutting quality, while the cutting time was chosen as a feature for processing efficiency. In the experiment, a steel plate of 20 mm length was subjected to straight cutting. Two pre-cut lines were made, and after cutting the first line, the temperature of the piercing point on the second line was measured using a temperature gun for validation of the heat transfer model. Simultaneously, the kerf width (KW) and heat-affected zone (HAZ) were measured, and the cutting time was calculated. These measurements were used to model the objective function of multi-objective optimization.

**Table 2.** Laser cutting parameters and their levels for experimental use.

| Symbols | Factors | Unit | Level 1 | Level 2 | Level 3 | Level 4 | Level 5 | Level 6 |
|---------|---------|------|---------|---------|---------|---------|---------|---------|
| $f$ | Repetition frequency | kHz | 500 | 700 | 900 | 1100 | 1300 | 1500 |
| $P$ | Average power | w | 500 | 700 | 900 | 1100 | 1300 | 1500 |
| $v$ | Scanning speed | mm/s | 10 | 20 | 30 | 40 | 50 | 60 |

In the experiment, to reduce the spacing between the plates, increase the production yield, and lower the costs, narrower kerf widths and higher dimensional accuracy are required during the cutting and air path processes. This is carried out to avoid part damage during cutting and prevent material from melting due to excessive laser power. Therefore, in this study, the maximum repetition frequency of the laser used was 1500 kHz, the maximum average power was 1500 W, and the maximum average cutting speed was 60 mm/s. Obtaining more minor power levels and heat-affected zones are essential quality characteristics of the laser cutting process. Additionally, high process efficiency is also desired. Figure 11 illustrates the schematic of the kerf width (KW) and heat-affected zone (HAZ), where both the metal and surface oxide layers vaporize within the kerf, and the HAZ is the region where thermal transfer evaporates the internal metal only. In this study, the outermost layer, where the surface oxide layer melts but solidifies, is not considered, as the interior metal is not exposed.

The laser cutting machine adopts the Fiber Laser Cutting Machine of BYSTRONIC (SHENZHEN, CHINA) LASER TECHNOLOGY CO., Ltd. (Model: D-Emergy2060FCC6000W). The experimental setup diagram and the experimental site are shown in Figure 12. The kerf width and heat-affected zone were observed and measured using an optical microscope (Model: KEYENCE (OSAKA, JAPAN) VHX5000). Three regions were selected on both sides of the kerf to measure the width of the heat-affected zone, and the average width was used as a measurement standard for characterizing the heat-affected zone, as shown in Figure 13. The temperature, kerf width, and heat-affected zone obtained for the 50 sets of process parameters, along with the cutting time calculated using Equation (15), are presented in Table 3. A smaller cutting time represents a higher processing efficiency (e).

$$t = \frac{d}{v} \tag{15}$$

**Table 3.** Experimental combinations and results.

| No | *f* (kHz) | *P* (W) | *V* (mm/s) | *T* (K) | KW (µm) | HAZ (µm) | *t* (s) |
|---|---|---|---|---|---|---|---|
| 1 | 500 | 1500 | 50 | 650 | 41 | 9 | 0.2 |
| 2 | 1100 | 700 | 10 | 754 | 22 | 6 | 1 |
| 3 | 700 | 900 | 10 | 630 | 30 | 7 | 1 |
| 4 | 1500 | 500 | 10 | 673 | 18 | 5 | 1 |
| 5 | 500 | 1100 | 10 | 602 | 26 | 5 | 1 |
| 6 | 1100 | 1100 | 30 | 856 | 63 | 12 | 0.33 |
| 7 | 500 | 700 | 30 | 704 | 56 | 7 | 0.33 |
| 8 | 1100 | 700 | 30 | 670 | 33 | 9 | 0.33 |
| 9 | 500 | 1300 | 50 | 807 | 83 | 12 | 0.2 |
| 10 | 700 | 700 | 30 | 700 | 26 | 8 | 0.33 |
| 11 | 1300 | 1500 | 40 | 830 | 55 | 22 | 0.25 |
| 12 | 900 | 1500 | 40 | 839 | 60 | 18 | 0.25 |
| 13 | 500 | 900 | 50 | 721 | 85 | 8 | 0.2 |
| 14 | 1500 | 700 | 30 | 632 | 20 | 7 | 0.33 |
| 15 | 1100 | 900 | 60 | 776 | 44 | 12 | 0.17 |
| 16 | 700 | 900 | 50 | 753 | 40 | 9 | 0.2 |
| 17 | 900 | 1300 | 40 | 768 | 72 | 14 | 0.25 |
| 18 | 1500 | 900 | 50 | 632 | 48 | 9 | 0.2 |
| 19 | 1100 | 1100 | 20 | 655 | 55 | 15 | 0.5 |
| 20 | 500 | 1100 | 20 | 631 | 35 | 12 | 0.5 |
| 21 | 900 | 1300 | 10 | 756 | 80 | 18 | 1 |
| 22 | 1300 | 1300 | 40 | 779 | 64 | 19 | 0.25 |
| 23 | 700 | 1100 | 20 | 687 | 45 | 12 | 0.5 |
| 24 | 900 | 1500 | 60 | 567 | 25 | 5 | 0.17 |
| 25 | 1100 | 1300 | 40 | 635 | 56 | 18 | 0.25 |
| 26 | 700 | 900 | 20 | 555 | 39 | 6 | 0.5 |
| 27 | 900 | 1100 | 20 | 654 | 62 | 12 | 0.5 |
| 28 | 1300 | 1100 | 20 | 656 | 52 | 16 | 0.5 |
| 29 | 500 | 1300 | 40 | 777 | 41 | 32 | 0.25 |
| 30 | 1500 | 1500 | 20 | 756 | 56 | 11 | 0.5 |
| 31 | 1100 | 1500 | 30 | 879 | 47 | 21 | 0.33 |
| 32 | 1300 | 900 | 40 | 643 | 42 | 8 | 0.25 |
| 33 | 900 | 900 | 60 | 629 | 52 | 9 | 0.17 |
| 34 | 1500 | 1300 | 40 | 667 | 72 | 13 | 0.25 |
| 35 | 500 | 1500 | 20 | 656 | 75 | 40 | 0.5 |
| 36 | 700 | 1100 | 30 | 566 | 74 | 17 | 0.33 |
| 37 | 900 | 700 | 30 | 627 | 30 | 6 | 0.33 |
| 38 | 1500 | 1500 | 20 | 785 | 83 | 15 | 0.5 |
| 39 | 700 | 1300 | 40 | 643 | 50 | 14 | 0.25 |
| 40 | 1300 | 900 | 60 | 532 | 25 | 13 | 0.17 |
| 41 | 1100 | 700 | 20 | 653 | 32 | 8 | 0.5 |
| 42 | 1300 | 1300 | 10 | 853 | 75 | 22 | 1 |
| 43 | 1300 | 500 | 50 | 590 | 49 | 9 | 0.2 |
| 44 | 900 | 700 | 10 | 585 | 42 | 4 | 1 |
| 45 | 1300 | 700 | 30 | 720 | 35 | 10 | 0.33 |
| 46 | 700 | 1100 | 60 | 820 | 67 | 22 | 0.17 |
| 47 | 1300 | 500 | 40 | 600 | 27 | 5 | 0.25 |
| 48 | 900 | 1500 | 50 | 675 | 27 | 5 | 0.2 |
| 49 | 1300 | 500 | 10 | 590 | 30 | 7 | 1 |
| 50 | 500 | 1300 | 40 | 550 | 26 | 6 | 0.25 |

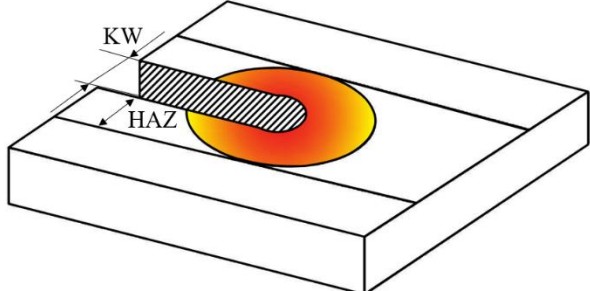

**Figure 11.** Schematic diagram of KW and HAZ.

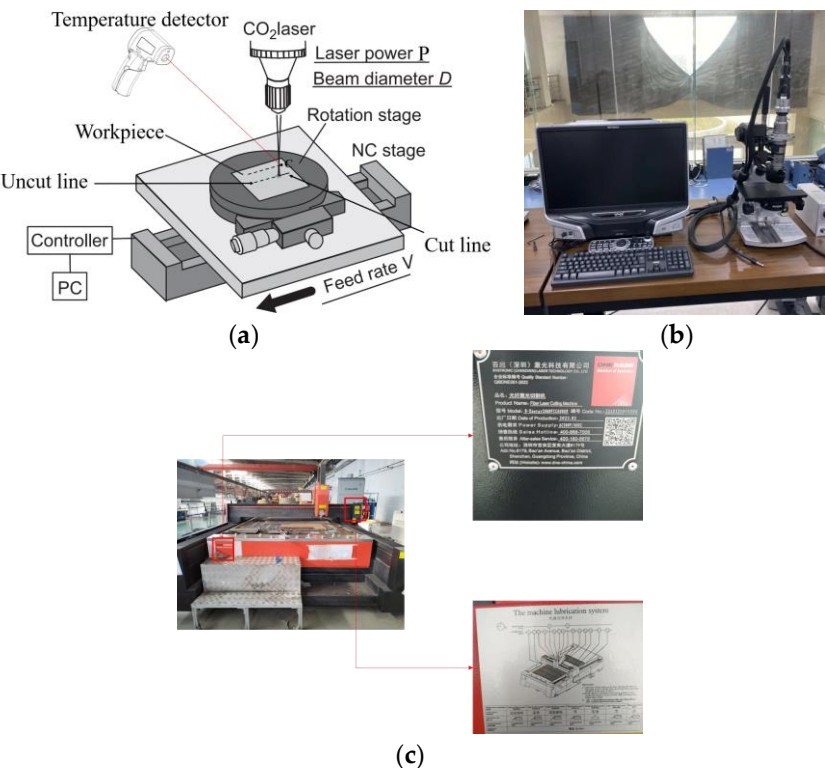

**Figure 12.** Schematic diagram of experimental setup. (**a**) Schematic diagram of experimental apparatus; (**b**) optical microscope; (**c**) laser cutting machine.

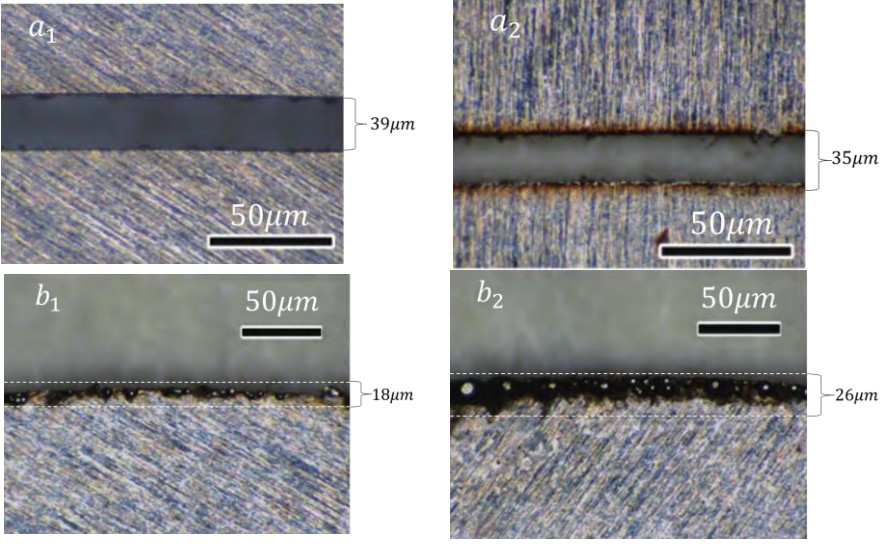

**Figure 13.** Observation images under an optical microscope. ($a_1$, $a_2$): kerf width measured under an optical microscope; ($b_1$, $b_2$): heat-affected zone measured under an optical microscope.

## 5. Results and Analysis

### 5.1. Analysis of Results of Heat Transfer Modeling

Based on the process parameter combinations in Table 1, $T_p(K)$ (predicted value) is calculated using the constructed heat transfer model. Using the actual temperature measurements under different combinations of process parameters in Table 3, the temperature curves shown in Figure 14, as well as the predicted curves, are plotted.

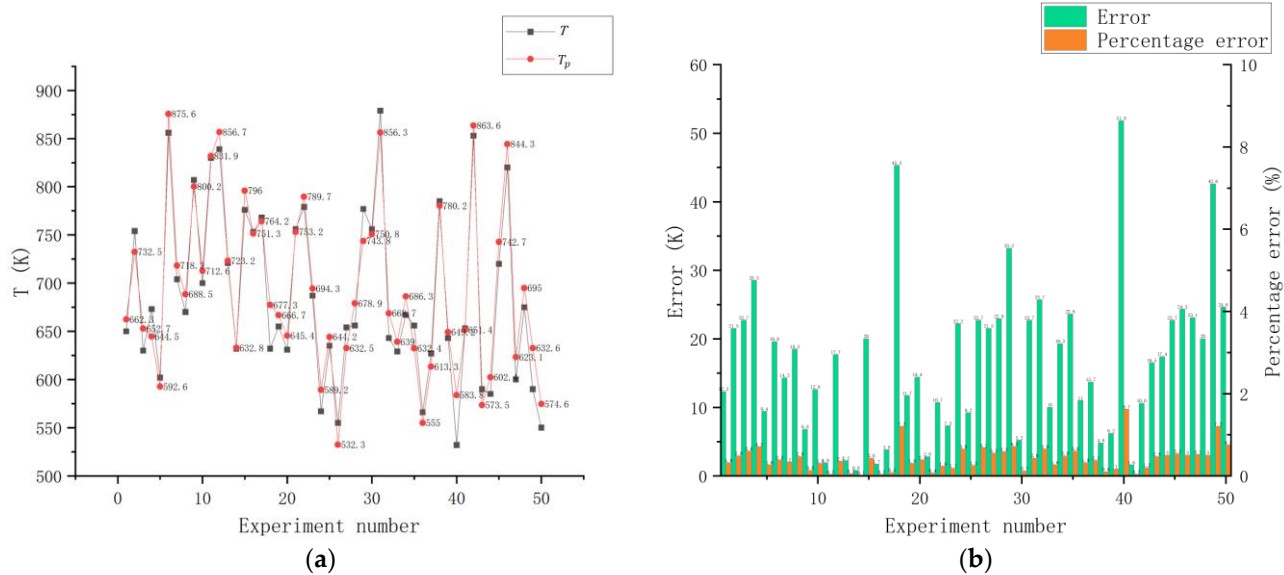

**Figure 14.** Heat impact prediction curves and model error analysis. (**a**) Predicted and measured values of temperature; (**b**) error analysis of heat transfer model.

From Figure 14a,b, it can be observed that the predicted values follow a similar trend as the measured values, and the error between the predicted values and the actual measurements is less than 10%. This confirms the effectiveness of the constructed heat transfer model.

### 5.2. Analysis of Results of Heat Transfer Modeling

To explore the relationship between process parameters and cutting quality features (kerf width, KW, and heat-affected zone, HAZ) and construct an optimization objective function, this study utilizes a backpropagation (BP) neural network to model the kerf width (KW) and heat-affected zone (HAZ). The BP neural network consists of an input layer, an output layer, and hidden layers, which adequately meet the requirements for accurate modeling of laser processing [31,32]. In this study, the structure of the BP neural network is illustrated in Figure 15. The input layer consists of three nodes corresponding to repetition frequency, average power, and scanning speed. The output layer consists of two nodes representing KW and HAZ. The number of nodes in the hidden layer is not predetermined and can influence the modeling accuracy of the BP neural network. Additionally, the selection of the activation function also impacts the model's accuracy. To determine the appropriate number of hidden layer nodes and activation functions to ensure modeling accuracy, this study employs particle swarm optimization to optimize the number of hidden layer nodes in the BP neural network and determine the activation functions [33]. All the activation function names and their codes are presented in Table 4.

**Table 4.** Activation function and its code.

| Activation Function | Tansig | Logsig | Elliotsig | Hardlim | Hardlims | Poslin | Purelin | Satlin |
| --- | --- | --- | --- | --- | --- | --- | --- | --- |
| Code | 0 | 1 | 2 | 3 | 4 | 5 | 6 | 7 |
| Activation functions | satlins | netinv | tribas | radbas | radbasn | compet | softmax | |
| Code | 8 | 9 | 10 | 11 | 12 | 13 | 14 | |

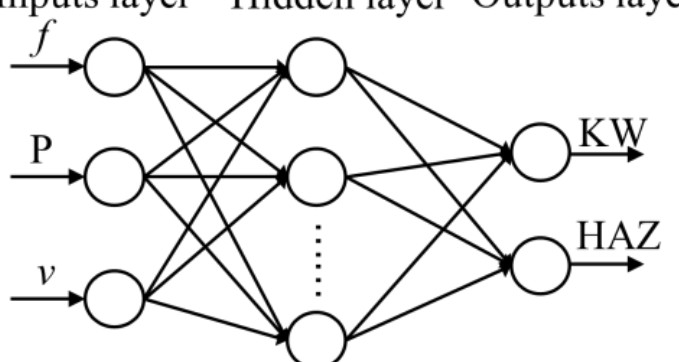

**Figure 15.** Structure of a BP neural network.

To quantitatively evaluate the model accuracy of PSO-BP, the coefficients of determination $MSE$ and $R^2$ are used as the evaluation indexes of the model accuracy. The calculation of $MSE$ and $R^2$ are shown in Equations (16) and (17) [34].

$$MSE = \frac{1}{m}\sum_{i=1}^{m}(\hat{y}_i - y_i)^2, \tag{16}$$

$$R^2 = 1 - \frac{\sum\limits_{i=1}^{m}(\hat{y}_i - y_i)^2}{\sum\limits_{i=1}^{m}(\overline{y}_i - y_i)^2}, \tag{17}$$

where $y_i$ is the experimental value, $\hat{y}_i$ is the predicted value, $\overline{y}_i$ is the mean value, and $m$ is the number of samples.

Using the 30 sets of data in Table 3 as the training set and the remaining 20 sets as the test set, in the process of optimizing BP using PSO, the PSO internal parameter settings are demonstrated in Table 5. The learning factors c1 and c2 represent the degree of contribution of the particle's local experience and the global optimal experience, respectively.

**Table 5.** Internal parameter list of PSO.

| Number of Particles (N) | Learning Factor | Particle Speed Range | Particle Position Range | Maximum Number of Iterations |
|---|---|---|---|---|
| 30 | c1 = c2 = 2 | −5~5 | −5~5 | 100 |

Based on the training of 30 sets of training sets, the learning rate of the neural network was set to 0.1, the number of nodes in the hidden layer of the BP neural network was determined by PSO, and the activation functions of the hidden and output layers adaptively selected by PSO are displayed in Table 6, where $a_1$ and $a_2$ denote the designations of the activation functions of the hidden and output layers, respectively.

As shown in Table 5, the optimal model is the PSO-BP model highlighted in bold in Table 6 with nine hidden nodes. The established models for HAZ(X) and KW(X) have $R^2$ values of 0.985 and 0.967, respectively, and MSE values of 115.27 $\mu m^2$ and 127.46 $\mu m^2$, which validate the effectiveness of the models. Based on the activation function designations in Table 4, the activation function for the hidden layer of PSO-BP is determined to be elliotsig, and the activation function for the output layer is logsig.

Based on the provided information, the accuracy of the PSO-BP model can be further evaluated using Figure 16. The accuracy of the model is reflected in the precision of its predictions for KW and HAZ. The model demonstrates a prediction error of less than 10% for both KW and HAZ, with the largest error of 9.2% observed for HAZ in the sixth

set of process parameters. This indicates that the established PSO-BP model exhibits high accuracy, and the optimized process parameters can meet the specified machining requirements, effectively improving the machining quality.

**Table 6.** Model performance comparison.

| Model | Number of Neurons | Transfer Function | | $R^2$ | | *MSE* | |
|---|---|---|---|---|---|---|---|
| | | $a_1$ | $a_2$ | KW | HAZ | KW $(\mu m^2)$ | HAZ $(\mu m^2)$ |
| PSO-BP | 9 | 3 | 2 | 0.932 | 0.974 | 132.67 | 123.56 |
| | **9** | **2** | **1** | **0.967** | **0.985** | **127.46** | **115.27** |
| | 9 | 4 | 1 | 0.945 | 0.943 | 124.35 | 108.45 |
| | 8 | 2 | 1 | 0.948 | 0.931 | 143.63 | 128.76 |
| | 8 | 2 | 1 | 0.953 | 0.894 | 153.27 | 132.32 |
| | 7 | 2 | 1 | 0.948 | 0.923 | 159.53 | 147.26 |
| | 7 | 3 | 1 | 0.955 | 0.847 | 164.43 | 163.23 |
| | 6 | 2 | 1 | 0.963 | 0.759 | 186.26 | 172.38 |
| | 6 | 2 | 1 | 0.943 | 0.832 | 203.26 | 198.28 |
| | 5 | 2 | 1 | 0.921 | 0.845 | 211.27 | 208.29 |
| | 5 | 2 | 1 | 0.893 | 0.844 | 212.38 | 217.29 |
| | 4 | 2 | 1 | 0.874 | 0.922 | 232.35 | 213.21 |
| | 4 | 2 | 1 | 0.854 | 0.873 | 222.36 | 232.27 |

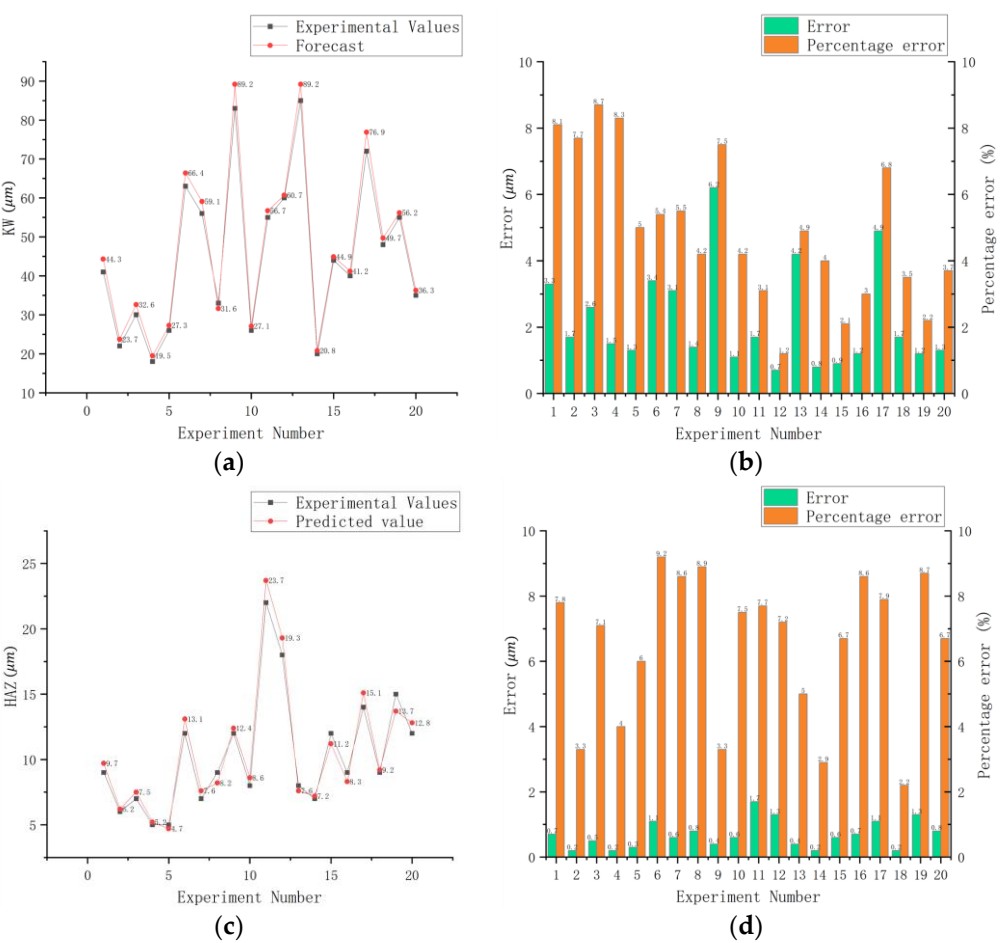

**Figure 16.** PSO-BP neural network model predicted and experimental values and model error analysis. (**a**) Predicted and experimental values of KW; (**b**) model error analysis of KW; (**c**) predicted and experimental values of HAZ; (**d**) model error analysis of HAZ.

Based on the established PSO-BP model, the effect of process parameters on KW and HAZ was analyzed, and the results are shown in Figure 17. Figure 17a illustrates the impact of process parameters on KW. It is evident that power has the most significant influence on KW, as KW linearly increases with power. The main reason for this is that as power increases, the material absorbs more laser energy per unit of time, leading to an accumulation of heat within the material and a greater increase in KW. As cutting speed increases, the distance between laser impact points increases, resulting in a decrease in accumulated energy per unit area and a decrease in KW. With an increase in pulse repetition rate, the pulse energy decreases, and due to the Gaussian distribution of laser beam energy, the high-energy interaction area is reduced, leading to a stabilization of KW.

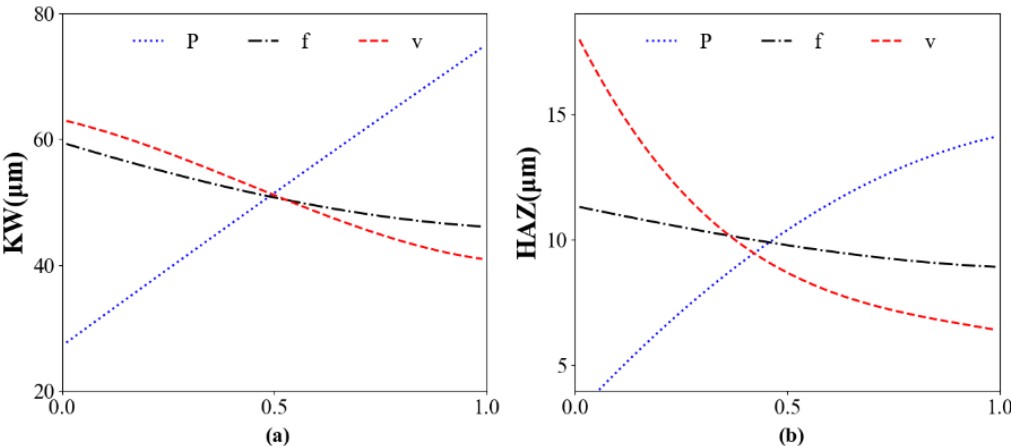

**Figure 17.** Influence of process parameters on machining quality. (**a**) Effect of process parameters on KW; (**b**) effect of process parameters on HAZ.

In Figure 17a,b, it can be observed that the influence of process parameters on HAZ follows a trend similar to the KW. The effect of cutting speed on HAZ is higher than its impact on KW. Additionally, the most significant factor affecting HAZ is the cutting speed.

*5.3. Solution Analysis of Multi-Objective Optimization Model for Machining Quality and Efficiency Based on NSGA-II*

A multi-objective optimization model (Equation (18)) for the quality and efficiency of the cutting process is constructed based on the PSO-BP-constructed models of KW and HAZ (KW(X) and HAZ(X) and the model of the processing efficiency feature t(x) computed using Equation (15), where X = [$f, P, V$].

$$\min Y(X = f, P, V) = \min(\text{HAZ}(X), \text{KW}(X), \text{t}(X))$$
$$\begin{cases} 500\,\text{kHz} \leq f \leq 1500\,\text{kHz} \\ 500\,\text{W} \leq P \leq 1500\,\text{W} \\ 10\,\text{mm/s} \leq V \leq 60\,\text{mm/s} \end{cases} \tag{18}$$

In laser processing, there is often a trade-off between process quality and efficiency. Solely minimizing KW and HAZ may lead to a decrease in process efficiency. The NSGA-II can address this issue by considering multiple optimization objectives and optimizing process parameters without compromising at least one other objective. The process of obtaining the Pareto front using NSGA-II, as described in Equation (18), is as follows:

Step 1: Set the population size and generate a random number of process parameter sets. In this study, the range of the three process parameters was limited based on Table 4. The population size was set to 500, and 500 sets of process parameters were randomly generated within the specified parameter ranges.

Step 2: Evaluate the non-dominated population and rank them. The process parameters were evaluated using the established PSO-BP model and formulas, and a non-

dominated sorting technique was applied. The optimization objectives were to minimize KW and HAZ while minimizing the processing time.

Step 3: Select parents from the population using binary tournament selection. The binary tournament selection (BTS) strategy was used to select the optimal process parameters from the population, forming the offspring for the next generation.

Step 4: Perform crossover and mutation operations on each generation and select offspring from the crossover and mutation operators. By applying crossover and mutation operations, new offspring were generated, resulting in a greater variety of process parameter combinations. The newly obtained process parameters were combined with the selected high-quality process parameters from Step 3, forming new offspring that were then transferred back to Step 2. This iterative process continued until the maximum number of iterations was reached, obtaining the optimal Pareto front.

The Pareto front obtained through multi-objective optimization using NSGA-II is shown in Figure 18. In the Pareto front shown in Figure 18, each point in the set of solutions represents a surface under the three optimization objectives. Optimizing one objective inevitably weakens the other objectives, and achieving a common optimal solution for all objectives is impossible.

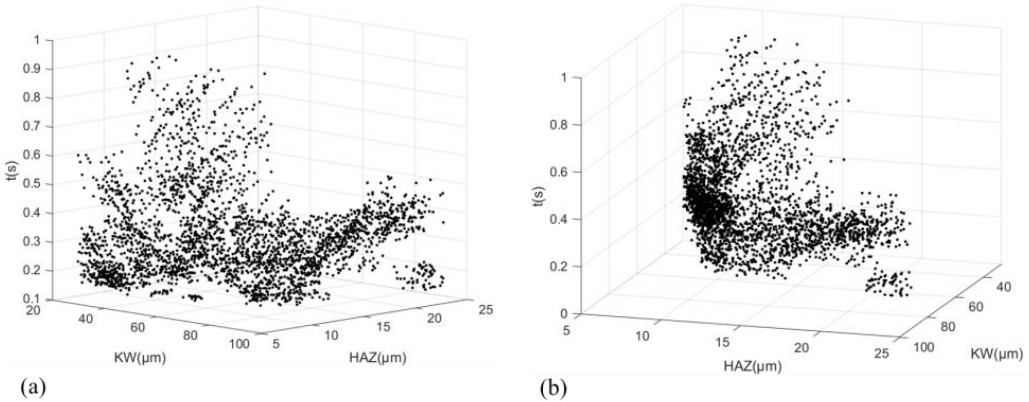

(a)  (b)

**Figure 18.** Pareto frontier after three-objective optimization. (**a**) Pareto front for three-objective optimization; (**b**) the Pareto front of the three-objective optimization in the side-looking direction.

To observe the trends of the Pareto front from different perspectives, the projection graphs of Figure 18 in three different directions are provided in Figure 19. Figure 19a shows the projection ignoring KW, with HAZ and t as the axes. The red curve represents the specific trend. It can be observed that in the initial stage, as HAZ increases, t decreases sharply and then decreases gradually. This indicates that in situations where high process efficiency is required, the variation in material processing quality is slow and has a minimal impact. Figure 19b shows the projection ignoring HAZ, with KW and t as the axes. The red curve represents the specific trend. It can be observed that in the initial stage, as KW increases, t decreases sharply and then decreases gradually. This again indicates that in situations where high process efficiency is required, the variation in material processing quality is slow and has a minimal impact. Figure 19c shows the projection ignoring t, with HAZ and KW as the axes. The red curve represents the specific trend. It can be observed that in the initial stage, as KW increases, HAZ decreases sharply and then rises sharply. This suggests that the two criteria for evaluating the processing quality have a relatively small mutual influence. These projections provide a focused view of the Pareto front, allowing for a better understanding of the trade-offs and relationships between the different optimization objectives.

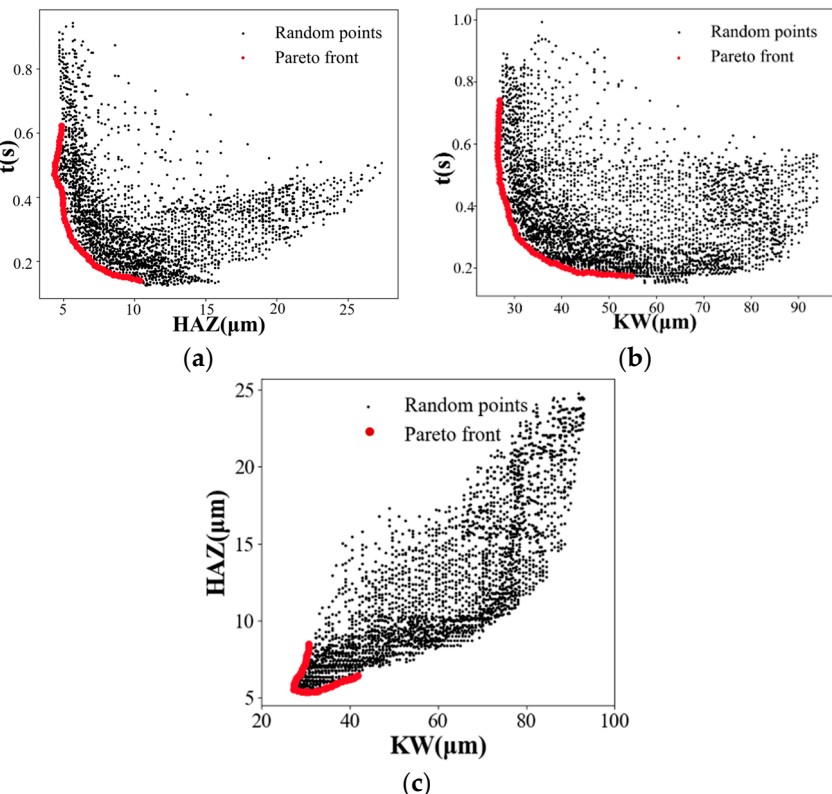

**Figure 19.** Projections of Pareto diagrams in three directions. (**a**) Effects between t and HAZ; (**b**) effects between t and KW; (**c**) effects between KW and HAZ.

*5.4. Integration of TOPSIS Decision-Making Methods and Temperature Control for Solving Process Parameter Combinations A and B*

From the analysis of the Pareto front solution set in the previous section, it is known that achieving better process quality may require sacrificing certain processing efficiency. Therefore, not all solutions in the Pareto front are suitable for practical processing. Conventional methods often use intervals with smooth trend changes as references for processing recommendations. However, this simple approach is crude and cannot explain how to obtain the optimal solution. The selection of the optimal solution can be based on the decision maker's needs or preferences, or it can be further refined through comprehensive evaluations. To obtain a solution that balances the desired comprehensive performance, this study utilizes the heat transfer model $T_{C_i}(r, t)$ to calculate the thermal influence value $T_p$ for the Pareto front solution set. Based on the set range limits, the parameter combinations are selected, and TOPSIS is used for further selection of the optimal solution.

In actual machining processes, parts are often processed in groups. During the cutting process, the ring-shaped heat sources generated by each part's cutting will affect the perforation point of the next part. As the number of processed parts increases, the thermal accumulation at the perforation point of the parts to be processed continuously increases, which affects the machining quality. Therefore, the selected machining parameters not only need to ensure corresponding guarantees for processing efficiency and quality but also need to ensure that there is no significant heat effect on the upcoming parts. By calculating the thermal influence value $T_p$ in the Pareto front solution set, temperature intervals can be defined. From these intervals, machining parameter combinations that meet the requirements for minimal thermal influence on the parts in the Pareto solution set can be selected.

Using the heat transfer model $T_{C_i}(r, t)$, the temperature values for 100 sets of Pareto front solutions can be calculated. Since the thermal influence is related to the perforation point in actual machining, and the orthogonal experimental process involves cutting a

straight line, we can assume that the next perforation point is located 6 mm away from the straight line (as the minimum distance between two parts in the group is 6 mm). Additionally, the perforation point on the straight line is perpendicular to the line connecting it to the assumed perforation point. Simulating the thermal accumulation temperature at this point, we can select machining parameter combinations from the results where the temperature is less than 700 K. The table below (Table 7) shows the simulated temperature values, with the underlined part of the parameter with a temperature greater than 700 K.

**Table 7.** Pareto front predicted temperature values.

| No | *f* (kHz) | *P* (W) | *V* (mm/s) | *T* (K) |
|----|-----------|---------|------------|---------|
| 1 | 500 | 1500 | 50 | 650 |
| 2 | ~~1100~~ | ~~700~~ | ~~10~~ | ~~754~~ |
| 3 | 800 | 900 | 10 | 630 |
| 4 | 1500 | 500 | 10 | 647 |
| 5 | 500 | 1100 | 10 | 602 |
| 6 | ~~1400~~ | ~~1100~~ | ~~45~~ | ~~856~~ |
| ⋮ | ⋮ | ⋮ | ⋮ | ⋮ |
| 95 | 1100 | 700 | 35 | 670 |
| 96 | ~~1450~~ | ~~1300~~ | ~~50~~ | ~~807~~ |
| 97 | 550 | 1100 | 20 | 631 |
| 98 | ~~950~~ | ~~1300~~ | ~~10~~ | ~~783~~ |
| 99 | ~~1200~~ | ~~1300~~ | ~~45~~ | ~~829~~ |
| 100 | 750 | 1100 | 20 | 687 |

The TOPSIS (Technique for Order of Preference by Similarity to Ideal Solution) method, first introduced by C.L. Hwang and K. Yoon in 1981 and further developed by Yoon [35] in 1987 and Hwang, Lai, and Liu [36] in 1993, is a technique for ranking evaluation objects based on their proximity to the idealized objectives. It provides a way to evaluate the relative superiority or inferiority of existing objects. In the case of the Pareto front solution set with 60 solutions, where each objective's response value varies significantly, simply selecting the minimum value for one objective may result in an increase in other objectives. The TOPSIS method can help address this issue by considering the overall performance of the solutions.

Using the TOPSIS method, two different sets of weights can be applied to filter out two sets of different process parameters, A and B. For the first set (Parameter A), the weight for HAZ is set to 0.3, the weight for KW is set to 0.3, and the weight for t is set to 0.4. This weight distribution emphasizes efficiency to ensure high-efficiency machining while also focusing on individual machining quality. However, due to the thermal accumulation phenomenon in the machining group, maintaining high efficiency can increase the impact of thermal accumulation. To address this issue, the second set of machining parameters (Parameter B) is designed. In this set, the weight for HAZ is set to 0.4, the weight for KW is set to 0.4, and the weight for t is set to 0.2. This weight distribution gives more importance to quality to prevent a decrease in quality caused by the continuous increase in thermal accumulation during the machining process. It allows temperature control using this set of parameters. The optimization results and TOPSIS values for the first group are presented in Table 8. The parameters highlighted in bold (Parameter A) represent the optimal machining parameters under the current conditions. These parameters allow for efficient machining while ensuring quality, provided that the temperature limit is not exceeded. Table 9 shows the optimization results and TOPSIS values for the second group. The parameters highlighted in bold (Parameter B) represent the optimal machining parameters under the current conditions. When the temperature limit is reached, this set of machining parameters can effectively reduce the temperature of the machined part and optimize the machining quality without sacrificing too much efficiency.

**Table 8.** TOPSIS optimization results (HAZ = 0.3, KW = 0.3, t = 0.4).

| | | Response Value | | TOPSIS | |
|---|---|---|---|---|---|
| **KW (μm)** | **HAZ (μm)** | **T (K)** | **t (s)** | **Score** | **Rank** |
| 33.9495 | 18.2772 | 622 | 0.1380 | 0.3797 | 23 |
| 49.7396 | 23.0633 | 753 | 0.0871 | 0.3108 | 46 |
| 36.5374 | 18.1650 | 654 | 0.0851 | 0.3069 | 39 |
| ⋮ | ⋮ | ⋮ | ⋮ | ⋮ | ⋮ |
| **21.0912** | **9.0486** | **703** | **0.0363** | **0.2342** | **1** |
| ⋮ | ⋮ | ⋮ | ⋮ | ⋮ | ⋮ |
| 39.7396 | 21.1343 | 781 | 0.0871 | 0.1927 | 22 |
| 34.5447 | 20.1650 | 763 | 0.0932 | 0.1850 | 15 |
| 49.5447 | 30.1650 | 832 | 0.1380 | 0.1738 | 56 |
| 53.9217 | 39.0486 | 853 | 0.1954 | 0.1322 | 49 |

**Table 9.** TOPSIS optimization results (HAZ = 0.4, KW = 0.4, t = 0.2).

| | | Response Value | | TOPSIS | |
|---|---|---|---|---|---|
| **KW (μm)** | **HAZ (μm)** | **T (K)** | **t (s)** | **Score** | **Rank** |
| 68.9479 | 33.5992 | 876 | 0.2331 | 0.3167 | 53 |
| 66.5347 | 33.6084 | 864 | 0.0871 | 0.2933 | 50 |
| 40.9435 | 27.0752 | 744 | 0.0851 | 0.2846 | 30 |
| ⋮ | ⋮ | ⋮ | ⋮ | ⋮ | ⋮ |
| **16.4685** | **6.3452** | **625** | **0.0721** | **0.2239** | **1** |
| ⋮ | ⋮ | ⋮ | ⋮ | ⋮ | ⋮ |
| 26.0921 | 13.6283 | 676 | 0.0731 | 0.1557 | 17 |
| 36.0921 | 15.0486 | 706 | 0.0532 | 0.1438 | 22 |
| 39.6876 | 17.0583 | 732 | 0.0518 | 0.1444 | 27 |
| 49.7396 | 21.1343 | 794 | 0.0598 | 0.1283 | 37 |

The critical temperature of the steel plate used in the experiment is 995 K, as shown in Table 1. When the critical temperature is reached, the material changes. Therefore, the set temperature must be significantly lower than the critical temperature. As adjusting the process parameters when reaching the temperature boundary does not immediately reduce the temperature below the boundary, the set temperature should be much lower than the critical temperature. Based on real-time temperature data obtained from high-quality machined parts during the experiment, the upper temperature limit is set to 750 K, and the lower temperature limit is set to 650 K.

When using the first set of machining parameters (Parameter A) for machining, the thermal accumulation at the perforation point of the workpiece increases as the number of machined parts increases. This results in a continuous temperature rise. When the temperature reaches or exceeds $T_1$ (in this case, chosen as 750 K based on experience), it is necessary to reduce thermal accumulation. Therefore, the second set of machining parameters (Parameter B) is selected to improve the machining quality. The temperature profile of the entire plate is shown in Figure 20. By switching between the two parameter sets during machining, the thermal influence on the next perforation point is controlled within the temperature range of 500–750 K. Through temperature monitoring at the perforation point, the machining parameters are adjusted to regulate the laser machining process.

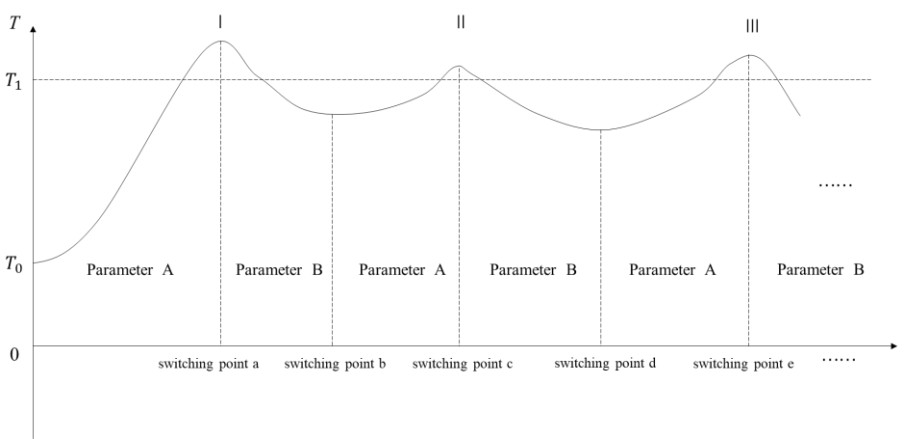

**Figure 20.** Sheet temperature control curve.

### 5.5. Simulation and Experimentation

To verify the optimization results, a machining comparison was conducted using the two sets of optimized parameters, and the results are summarized in Table 10. From the data in the table, it can be observed that the prediction errors for the entire machining process are all below 10%.

**Table 10.** Process parameter prediction and validation.

| | Processing Parameters | | | Predictions | | | Experiments | | | |
|---|---|---|---|---|---|---|---|---|---|---|
| | $f$ (kHz) | $P$ (w) | $V$ (mm/s) | KW (μm) | HAZ (μm) | t | KW (μm) | HAZ (μm) | t | Error |
| a | 1260 | 1075 | 45 | 21.0912 | 9.0486 | 0.222 | 22.5 | 9.8 | 0.215 | 8.3% |
| b | 1050 | 850 | 30 | 16.4685 | 6.3452 | 0.333 | 17.2 | 6.5 | 0.337 | 9.6% |

Before performing parameter optimization, it is necessary to determine the machining path. In order to ensure the feasibility of the simulation, this paper refers to the work of Makbul Hajad [12] on thermal-influenced machining path optimization. The principle for determining this path is incorporating thermal constraints into the calculations. When the temperature in a region of the workpiece exceeds a critical level, that region must be removed from the list of possible perforation points. This ensures the selection of an optimal path.

Based on the paths generated using four different initial machining parameters in the paper [12], this study selects two sets of different initial machining parameters for optimization. The machining parameters provided in the paper are used as the initial parameters. Using the thermal transfer model, the temperature at the pre-machining perforation points is simulated. The TOPSIS optimization results are then used to control the temperature and keep the thermal influence on the next perforation point within the temperature range of 500–750 K. The paths corresponding to different initial machining parameters are shown in Figures 21a and 23a. The comparison between constant machining parameters and optimized process parameters is presented in Tables 11 and 12. The control curves for different parameters and paths are shown in Figures 22 and 24.

Equation (19) represents the total laser cutting processing time, where t represents the total time, $t_{cutting}$ represents the laser cutting time, $t_s$ represents the tool start–stop time and laser perforation time, and $t_{air}$ represents the tool's air path time during the processing. The total laser cutting processing time comprises the laser cutting time, tool start–stop time, and air path time. Due to different tool paths under different processing parameters, the total laser cutting time is not the same. The numerical improvement in machining quality is obtained by comparing the kerf width and the average value of the heat-affected zone with their respective values under constant parameters and selecting the smaller improvement

value as the quality improvement value. The efficiency improvement value is the sum of the machining and cutting time in the table, combined with the tool start–stop time and the tool air path time in the actual processing, and finally, the efficiency increase is obtained. When the initial laser power is 500 W, and the line speed is 10 mm/s, the actual tool start–stop time and air path time during processing are 90 s. Compared to before optimization, the quality improvement after optimization is 8.63%, and the processing efficiency is increased by 20.6%, as shown in Table 11. When the initial laser power is 1500 W, and the line speed is 10 mm/s, the actual tool start–stop time and air path time during processing are 120 s. Compared to before optimization, the quality improvement after optimization is 14.53%, and the processing efficiency is increased by 15.1%, as shown in Table 12.

**Table 11.** Comparison of results before and after optimization (laser power = 500 W; line speed = 10 mm/s).

| Processing Type | Number of Pieces Processed | $f$ (kHz) | $P$ (w) | $v$ (mm/s) | T (K) | KW (μm) | HAZ (μm) | Quality Improvement | t (s) | Efficiency Improvement |
|---|---|---|---|---|---|---|---|---|---|---|
| Constant parameter processing | | 500 | 500 | 10 | 704 | 28.7546 | 17.3586 | | 35 | |
| Optimized parametric machining | 1 | 500 | 500 | 10 | 532 | 17.6452 | 7.8457 | | 1 | |
| | 2 | 1260 | 1075 | 45 | 565 | 22.3569 | 9.8423 | | 0.222 | |
| | ⋮ | ⋮ | ⋮ | ⋮ | ⋮ | ⋮ | ⋮ | 8.63% | ⋮ | 20.6% |
| | 16 | 1260 | 1075 | 45 | 776 | 24.9345 | 14.3352 | | 0.222 | |
| | 17 | 1050 | 850 | 30 | 754 | 24.6353 | 13.7356 | | 0.333 | |
| | ⋮ | ⋮ | ⋮ | ⋮ | ⋮ | ⋮ | ⋮ | | ⋮ | |
| | 34 | 1260 | 1075 | 45 | 712 | 29.7618 | 17.7367 | | 0.222 | |
| | 35 | 1260 | 1075 | 45 | | 30.5342 | 18.7634 | | 0.333 | |

**Table 12.** Comparison of results before and after optimization (laser power = 1500 W; line speed = 10 mm/s).

| Processing Type | Number of Pieces Processed | $f$ (kHz) | $P$ (w) | $V$ (mm/s) | T (K) | KW (μm) | HAZ (μm) | Quality Improvement | t (s) | Efficiency Improvement |
|---|---|---|---|---|---|---|---|---|---|---|
| Constant parameter processing | | 1000 | 1500 | 10 | 756 | 31.6784 | 22.0937 | | 35 | |
| Optimized parametric machining | 1 | 1000 | 1500 | 10 | 579 | 23.5243 | 11.5443 | | 1 | |
| | 2 | 1260 | 1075 | 45 | 596 | 21.9096 | 9.2941 | | 0.222 | |
| | ⋮ | ⋮ | ⋮ | ⋮ | ⋮ | ⋮ | ⋮ | 14.53% | ⋮ | 15.1% |
| | 11 | 1260 | 1075 | 45 | 794 | 27.4537 | 17.5635 | | 0.222 | |
| | 12 | 1050 | 850 | 30 | 773 | 25.5321 | 16.6245 | | 0.333 | |
| | ⋮ | ⋮ | ⋮ | ⋮ | ⋮ | ⋮ | ⋮ | | ⋮ | |
| | 34 | 1050 | 850 | 30 | 759 | 32.5367 | 22.3633 | | 0.333 | |
| | 35 | 1260 | 1075 | 45 | | 30.5342 | 24.5237 | | 0.222 | |

$$t = t_{cutting} + t_s + t_{air} \tag{19}$$

The processing parameters were simulated, respectively, when the laser power was 500 W, 1000 W, and 1500 W and the cutting speed was 10 mm/s and 50 mm/s. The maximum value of the ordinate corresponds to 1500 K, and the simulation results are shown in Figure 25. The average maximum temperature is reduced by 15.3%. This indicates that the heat accumulation in laser cutting of the workpiece is reduced, leading to improved workpiece quality and increased efficiency while ensuring workpiece quality.

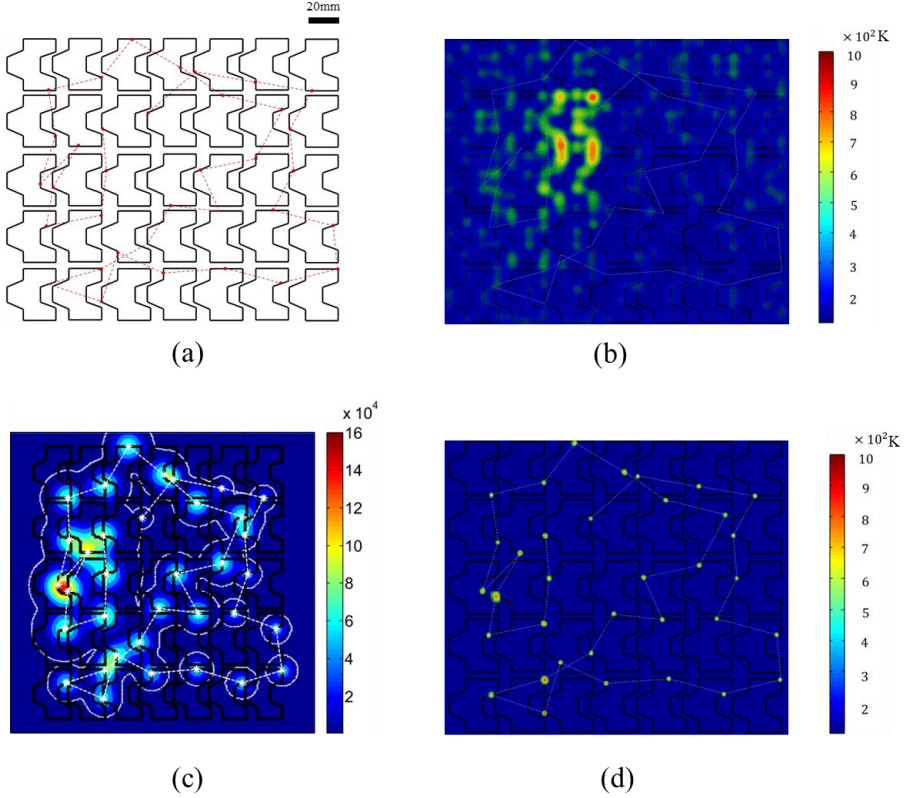

(a)  (b)

(c)  (d)

**Figure 21.** (**a**) Processing route after path optimization (laser power = 500 W; line speed = 10 mm/s); (**b**) thermogram of the entire workpiece after laser cutting; (**c**) heat map of punched points machined with constant machining parameters after path optimization [12]; (**d**) heat map of perforation point after parameter optimization.

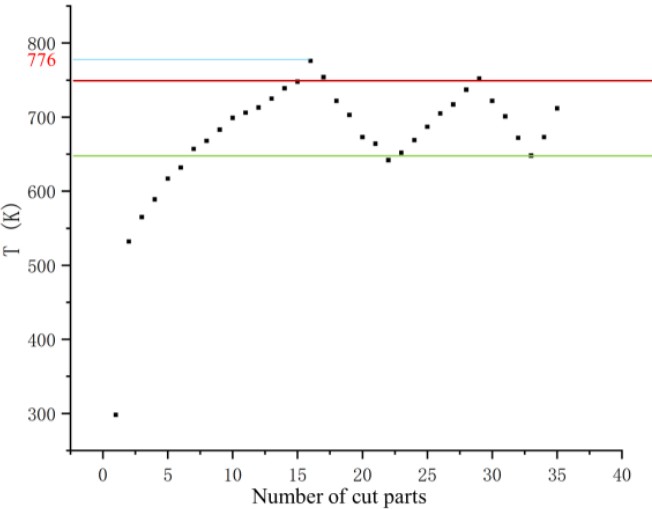

**Figure 22.** Regulation curve at laser power = 500 W, line speed = 10 mm/s.

Laser machining experiments were conducted on the pre-optimized and post-optimized machining parameters to verify the optimization effectiveness of the proposed method. The contour diagrams of the cut parts are shown in Figure 26, where Figure 26a,c represent the parts obtained with the optimized process parameters, while Figure 26b,d represent the parts obtained without optimization. It can be observed that the parts obtained without optimization have more burrs on the surface.

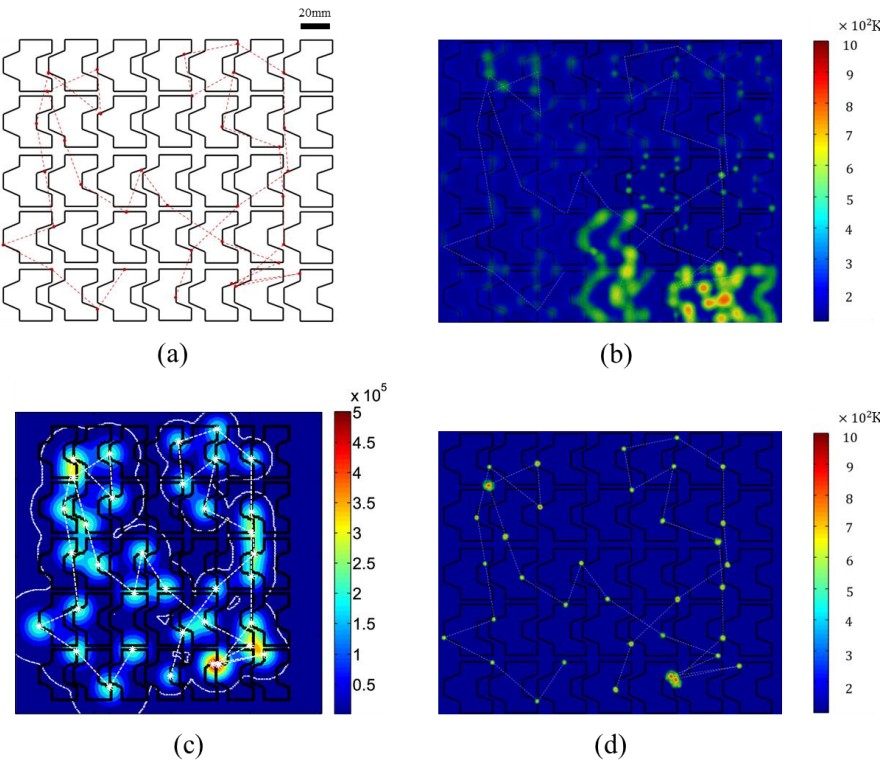

**Figure 23.** (**a**) Processing route after path optimization (laser power = 1500 W; line speed = 10 mm/s); (**b**) thermogram of the entire workpiece after laser cutting; (**c**) heat map of punched points machined with constant machining parameters after path optimization [12]; (**d**) heat map of perforation point after parameter optimization.

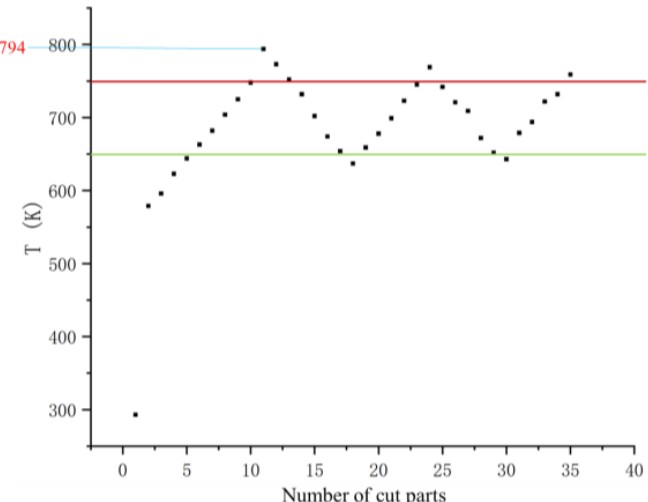

**Figure 24.** Regulation curve at laser power = 1500 W, line speed = 10 mm/s.

The thermal affected zone and surface smoothness are shown in Figure 27. The parts obtained with the optimized process parameters exhibit a smaller thermal affected zone and a smoother surface. In Figure 27b, there is almost no exposed metal debris. The optimized combination of machining parameters results in better machining quality, thus validating the effectiveness of the optimization method.

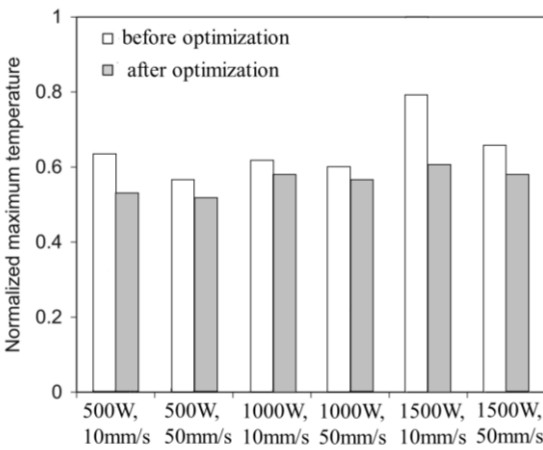

**Figure 25.** Normalized maximum temperature of the workpiece under different conditions.

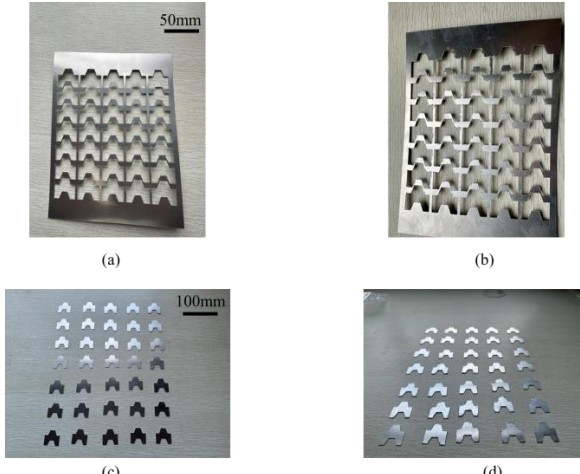

**Figure 26.** Comparison of experimental results before and after optimization. (**a**) Optimized cut sheet diagram; (**b**) before optimization diagram of cutting sheet; (**c**) optimized part group drawing; (**d**) part group drawing before optimization.

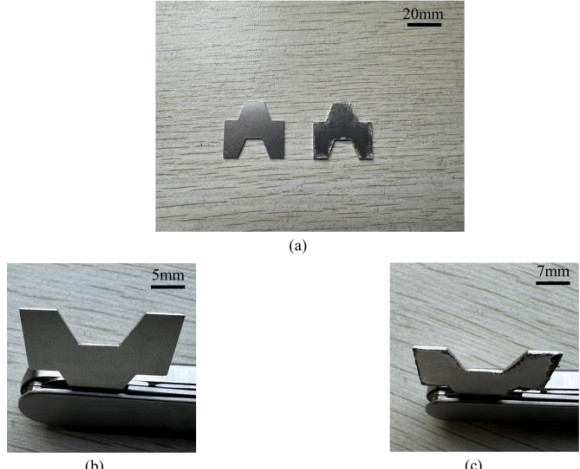

**Figure 27.** Comparison of individual parts. (**a**) Comparison of surface heat-affected zone and surface roughness before and after optimization; (**b**) heat-affected zone and surface roughness of the optimized cut area; (**c**) heat-affected zone and surface roughness of the cutting area before optimization.

## 6. Conclusions

This study addresses the issue of thermal accumulation in laser cutting of thin-walled sheet metal parts under the constraint of the shortest path. A segmented optimization method considering thermal influence is proposed. It involves predicting the temperature at the piercing points, solving the multi-objective optimization of quality and efficiency, setting weights for quality and efficiency in segmented stages based on the temperature at the piercing points, and utilizing a decision-making method that incorporates the temperature at the perforation points. This approach achieves improved quality and efficiency in the machining process, providing guidance and methods for practical machining. The main research conclusions are as follows:

(1) A thermal transfer model is constructed to predict the temperature at the perforation points during the cutting process by employing physical and mathematical modeling and analyzing the heat diffusion during cutting. The prediction model's error is less than 10%, verifying its effectiveness. This provides reliable technical support for temperature prediction in segmented control processes, ensuring precise temperature regulation during the cutting process.

(2) A prediction model for machining quality features (kerf width and heat-affected zone) is established using the PSO-BP method. Based on training and testing with experimental data, the developed models for HAZ(X) and KW(X) achieved $R^2$ values of 0.985 and 0.967, respectively. The corresponding mean squared error (MSE) values were 115.27 $\mu m^2$ and 127.46 $\mu m^2$. Additionally, the prediction errors for both KW and HAZ were found to be less than 10%. Notably, the largest prediction error for HAZ occurred in the sixth set of process parameters, with an error of 9.2%. These results demonstrate the high accuracy of the established PSO-BP model, indicating that the optimized process parameters can meet the specified machining requirements and effectively enhance the overall machining quality.

(3) Using the TOPSIS decision-making method, during the cutting stages where the temperature was below 650 K and above 750 K, separate weights were assigned to prioritize efficiency and quality, respectively. The decision-making process was constrained by the temperature range of 500 K–750 K. Two sets of process parameter combinations, A = [1260 kHz, 1075 w, 45 mm/s] and B = [1050 kHz, 850 w, 30 mm/s], were selected based on the criteria of high efficiency and high quality.

(4) The heat transfer model was used to predict the temperature at the perforation point for a group of sheet metal parts using two sets of process parameters. The predicted temperatures for the first set (Table 11) and the second set (Table 12) exceeded 750 K at the 16th and 22nd parts, respectively, during the validation experiments. Process parameter combination B was chosen for further machining. By adjusting the parameters, the temperatures for both sets fell within the specified range. As a result, the quality of the parts improved by 14.5%, and efficiency increased by 20.6%. These results validate the effectiveness of the proposed segmented optimization method.

However, there are some limitations in this study. In the laser machining process, this research adjusts the temperatures by finding relatively optimal sets of process parameters. However, real-time generation of appropriate machining parameters to avoid thermal influence can further enhance both efficiency and quality. Therefore, in future work, temperature impact will be included as a constraint in multi-objective optimization, and a database will be established to improve the model's accuracy.

**Author Contributions:** All authors contributed to the study's conception and design. Y.W. is mainly responsible for literature retrieval, experimental design, data analysis, model establishment, and manuscript writing. J.L. and J.M. are primarily responsible for directing the content and structure of the thesis and reviewing and revising the manuscript. X.L. provided guidance on research methodology and direction while reviewing the manuscript. All authors have read and agreed to the published version of the manuscript.

**Funding:** This research is supported by the National Natural Science Foundation of China (NSFC) (Grant No. 52165062), the Guangxi Natural Science Foundation Program (Grant No. 2020JJD160004), and the 2023 Guangxi Graduate Education Innovation Plan Project (Grant No. A3010022005).

**Data Availability Statement:** Data are contained within the article.

**Conflicts of Interest:** The authors declare no conflicts of interest.

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
