# Peer review of "Optimization Method of Sheet Metal Laser Cutting Process Parameters under Heat Influence"

_machines, doi:10.3390/machines12030206_

Round 1

Reviewer 1 Report

Comments and Suggestions for Authors

This work proposes a segmented optimization method for the laser-cutting process conditions. It is interesting, but some issues should be noted.

1.     The authors need to further optimize the structure of this work due to the poor logic.

2.     The authors must double-check the article's writing. Lots of grammatical mistakes were found.

3.     The authors need to supplement the relevant literature further in the introduction. For example, there are only 10 articles from Line 39 to Line 125. In addition, the contribution of this research should be claimed in the conclusion rather than the introduction.

4.     Please explain how to alternate the parameters of A and B in laser cutting. Does the laser system change the parameters of A and B automatically? If not, the loss of ablation energy is considered?

5.     Please provide the basis or reference for setting the upper and lower temperature limits (T1 and T2) in this article.

6.     In Fig. 13, please provide the predicted values of TP.

7.     How are the improvements of 14.53% in quality and 20.6% in efficiency determined?

Comments on the Quality of English Language

Moderate editing of English language required.

Reviewer 2 Report

Comments and Suggestions for Authors

This paper discusses the problems of workpiece deformation and excessive material melting caused by heat accumulation during laser cutting of thin plates, and proposes a segmented optimization method of laser cutting process parameters under thermal influence, which improves the processing quality and cutting efficiency to a certain extent by multi-objective optimization and increasing the temperature range as a constraint. It has certain innovation and large workload, but there are still the following problems to be pointed out:

1. The abstract section is too much and must be streamlined, in which the experiments and models of how to carry out the experiments do not need to be introduced, and it is directly stated what means are used and what kind of results are obtained, which needs to be reduced by at least half.

2.In the introduction, the author should explain the difference from other scholars' research and elaborate on the focus of this work. It is strongly suggested that the references need to make in-depth comments on the content of the cited papers; avoid generic comments. Mention/comment the relevance of the cited paper and especially the research gap associated to it. In addition, there are more relevant papers that should be covered in literature review:

Recent progress of chatter prediction, detection and suppression in milling

Chatter detection in milling process based on VMD and energy entropy

Towards high milling accuracy of Turbine Blades: A Review

3. Introduction at the end of the paper's contribution 1, 2, 3, the same is not streamlined enough, the main purpose is to improve the quality and efficiency of laser processing, so what is the specific value? Also some distinction needs to be made with the abstract and conclusion.

4. In the body of 79 lines, Figure 1 seems to be a citation of others, it should be indicated specifically which scholar, in addition, what is the unit of color labeling on the right side of Figure 1? What is the significance? Is it temperature or stress?

5. Scale is missing in Fig. 2, is there a citation for Eqs. 1-6? Or are they formulas derived by the authors?

6. What guidance can Equation 14 provide for the following? Does Figure 8 need a sub-figure number? The testing equipment in Figure 12 does not need to be described.

7. What is the neural network prediction accuracy in line 544? What is its average error? What is the maximum error? When was it obtained, this is key.

8. What are the scale units in Figs. 20, 22, and is a scale needed in Fig. 25? Pay attention to graphing normality.

9. Concise conclusions, as shown in question 1.

Comments on the Quality of English Language

Relatively high quality of English.

Reviewer 3 Report

Comments and Suggestions for Authors

The article “Optimization method of sheet metal laser cutting process parameters under heat influence” presents an interesting approach to optimizing the cutting process with a laser cutter. The title is appropriate to the content. The abstract is developed correctly. The introduction to the topic is sufficient. The methodology, reasoning and proceedings are presented in rather detail, which I consider an advantage. The study is quite extensive, so the authors have not avoided some errors, mainly of an editorial nature, but in some points information of merit nature should also be supplemented:

1. If the figure consists of several subfigures a), b), etc., all subfigures should be described in the caption (Figures 1, 17, 25).

2. Rows 134-135: What was the intention of the authors? RSM can also be done in non-commercial programs in Python or R, for example. For ANN, on the other hand, some kind of tool should also be used.

3. Row 168 - PSO-BP appears in the main text for the first time - the acronym needs to be clarified. Similarly NSGA-II in line 173, TOPSIS in line 176.

4. Section 2 - write down which programs were used at each stage during the study.

5. Figure 3 - the font on the graph is too small.

6. Row 299: a sentence cannot start with a variable symbol.

7. Row 313: unnecessary subsection.

8. Row 356, equation 8: the x sign is reserved for the vector product.

9. Row 393 - incorrect citation of Figs 9 and 10 (should be 6 and 7).

10. Row 413: repetition of "Each loop has the vertices we have defined earlier".

11. row 416: it is not clear what the subject of this sentence is.

12. Figures 8 and 9 are not referred to in the text.

13) Row 429 - please give the name/symbol of the steel.

14 Row 440 - what software was used?

15. The methodology for measuring KW and HAZ should be described in more detail: give the measurement parameters, the name of the measuring device, an example view of the measured sample under the microscope with the determined parameters.

16. Laser cutting machine - specify manufacturer and type.

17. Row 522 - instead of 'number of times', experimental value and predicted value would be more appropriate.

18. Figure 13b - convert the graph to 2D, and give units on the vertical axis. Units will be different - better to use two separate vertical scales - left and right.

19. Table 6 is missing units for MSE.

20. Row 558: “However, in Figure 16(b), the impact of power on HAZ is lower than its impact on KW”. This does not appear to be a valid statement. Both KW and HAZ change with P by about 300%.

21. Rows 483-485 and 604-606 - redundant text.

22 Table 7 - explain what strikethroughs are, give unit for V

23 Table 8 and 9 - explain what means is, give unit for T and t.

24. What is the temperature in Figure 19? Is it the maximum temperature of the plate?

25. Rows 729-731: ensure the correct order of figure’s quotation in the text.

26. Table 11 - give the units of the variables.

27. Figure 21 - What is the temperature in the figure? Is it the maximum temperature of the plate?

28. Figure 24 - no citation in the text. What was the reference value for computing normalized values?

29. Conclusion (4)- units should be rounded to significant places (due to measurement uncertainty). It is worth presenting the quoted information earlier in the Results.

Comments on the Quality of English Language

Minor editing

Round 2

Reviewer 1 Report

Comments and Suggestions for Authors

No comment.

Comments on the Quality of English Language

Minor editing of the English language required